# Extension of Operational Matrix Technique for the Solution of Nonlinear System of Caputo Fractional Differential Equations Subjected to Integral Type Boundary Constrains

**DOI:** 10.3390/e23091154

**Published:** 2021-09-02

**Authors:** Hammad Khalil, Murad Khalil, Ishak Hashim, Praveen Agarwal

**Affiliations:** 1Department of Mathematics, Attock Campus, University of Education, Lahore 43600, Pakistan; 2Department of Basic Sciences, University of Engineering and Technology Peshawar, Peshawar 25000, Pakistan; murad_khalil@uetpeshawar.edu.pk; 3School of Mathematical Sciences, Universiti Kebangsaan Malaysia, Bangi Selangor 43600, Malaysia; ishak_h@ukm.edu.my; 4Department of Mathematics, Anand International College of Engineering, Jaipur 303012, India; praveen.agarwal@anandice.ac.in

**Keywords:** approximation, numerical simulation, iterative methods

## Abstract

We extend the operational matrices technique to design a spectral solution of nonlinear fractional differential equations (FDEs). The derivative is considered in the Caputo sense. The coupled system of two FDEs is considered, subjected to more generalized integral type conditions. The basis of our approach is the most simple orthogonal polynomials. Several new matrices are derived that have strong applications in the development of computational scheme. The scheme presented in this article is able to convert nonlinear coupled system of FDEs to an equivalent S-lvester type algebraic equation. The solution of the algebraic structure is constructed by converting the system into a complex Schur form. After conversion, the solution of the resultant triangular system is obtained and transformed back to construct the solution of algebraic structure. The solution of the matrix equation is used to construct the solution of the related nonlinear system of FDEs. The convergence of the proposed method is investigated analytically and verified experimentally through a wide variety of test problems.

## 1. Introduction

Fractional calculus has a long and rich history. Its wide range of applications made this field an active area of mathematical research. Frequently investigated properties of FDEs include existence and uniqueness problems, multiplicity of solutions, stability of solutions and analytical study of the analytical properties of solution. Parallel to these area of research, one of the most explored and interested areas of research in this field is the design of new numerical methods for finding the approximate solutions to problems of this category. Many scientists and mathematicians are trying to design efficient and reliable techniques to find possible estimates to solutions of FDEs or their coupled systems.

There are many analytical, semi-analytical, numerical, and spectral approximations of solution to FDEs and their coupled systems. Among others, one of the easiest method is the differential transformation method (DTM). In [1], DTM is successfully applied to solve simple nonlinear FDEs with simple initial conditions. In [2], DTM is designed to solve the fractional-order counterpart of Korteweg De Vries (KDV) equation. The method is further improved for the solution of fractional-order boundary value problems in [3]. Solutions to fractional-order boundary value problems are also attempted with analytical methods such as the homotopy perturbation method; see for example [4,5,6]. The Adomian decomposition method is also a powerful analytical method [7,8,9]. Spectral methods have gained the attention of scholars in recent decades. Compared to other methods, spectral methods are easy to design and implement [10,11,12,13,14,15,16,17,18,19,20]. The availability of a wide range of orthogonal polynomials makes this method more interesting. They have the ability to solve fractional order problems, whose solutions are difficult or sometimes impossible to obtain with other traditional methods. For new readers, we strongly recommend studying the results obtained in [21,22,23,24,25,26] for a clear understanding and developing a good base. However, to the best of our knowledge, the spectral method becomes difficult and sometimes fails to handle the situation when boundary conditions are given in more complicated forms such as local conditions, nonlocal m-point terminal conditions, integral type terminal conditions, and radiation boundary conditions. Such conditions have solid application in various problems of traveling waves, heat conduction and electromagnetism. One can find a good example of application of integral type boundary condition to heat conduction phenomena in a rod of fixed length in a recent article [27].

Nonlocal FDEs arise in mathematical modeling of various problems in physics, engineering, ecology, and biological sciences [28,29,30]. Some of the numerical investigations regarding FDEs with nonlocal constrains are discussed in [31,32,33,34,35]. Numerical approaches such as finite difference and radial base function also remain a focus of interest. Application of these methods to one-dimensional heat-like equations has been studied in [32,36,37,38]. Two-dimensional diffusion problems [33,39,40] and Laplace equations with integral constraints are explored in [31].

Keeping in view the increasing interest of mathematicians in fractional calculus, we are strongly motivated to design a new spectral approximation technique for complicated problems such as
(1)$$\begin{array}{cc} \; & {}^{c}D^{\sigma_{1}}u\left(t\right)=f\left(u,v,u^{\left(\gamma_{1}\right)},v^{\left(\gamma_{2}\right)}\right),\\ \; & {}^{c}D^{\sigma_{2}}v\left(t\right)=g\left(u,v,u^{\left(\gamma_{1}\right)},v^{\left(\gamma_{2}\right)}\right),\\ \; & u\left(0\right)=u_{0},\;u\left(\tau \right)=m_{1}\int_{0}^{\omega_{1}}s\left(t\right)u\left(t\right)dt,\;0<\omega_{1}\leq \tau ,\\ \; & v\left(0\right)=v_{0},\;v\left(\tau \right)=m_{2}\int_{0}^{\omega_{2}}r\left(t\right)v\left(t\right)dt,\;0<\omega_{2}\leq \tau . \end{array}$$
where $0<\gamma_{1},\gamma_{2}\leq 1,$
$1<\sigma_{1},\sigma_{2}\leq 2,$ t∈[0,τ]. The scalar functions u(t) and v(t) are the solution to be determined. *f* and *g* are nonlinear functions of u(t),
v(t) and its fractional order derivatives and is assumed to be in such a form that the solution of the problem exists.

We start our discussion by introducing some definitions and preliminary results.

## 2. Preliminaries

The following definitions and notations are important for our further analysis. More details and theoretical understanding of these results, see [41,42,43,44,45].

**Definition** **1.**
*The Riemann–Liouville α−order integral of $\phi \in (L^{1}\left[a,b\right],R)$ is defined by the following relation.*

$$I^{\alpha }\phi \left(t\right)=\frac{1}{\Gamma \left(\alpha \right)}\int_{a}^{t}{\left(t-s\right)}^{\alpha -1}\phi \left(s\right)ds,$$



**Definition** **2.**
*For $\phi \left(t\right)\in C^{n}\left[a,b\right]$, the α order Caputo derivative is defined as*

$$\begin{array}{cc} {\;}^{c}D^{\alpha }\phi \left(t\right)= & \frac{1}{\Gamma (n-\alpha )}\int_{a}^{t}\frac{\phi^{\left(n\right)}\left(s\right)}{{\left(t-s\right)}^{\alpha +1-n}}ds,\\ \; & n-1\leq \alpha <n\;,\;n\in N, \end{array}$$

*where n=[α]+1.*


From the above definition, it is easy to extract $I^{\alpha }t^{b}=\frac{\Gamma (1+b)}{\Gamma (1+b+\alpha )}t^{b+\alpha }\;\mathrm{for}\;\alpha >0,\;k\geq 0$, ${}^{c}D^{\alpha }C=0,\;$ and
(2)$${}^{c}D^{\alpha }t^{b}=\frac{\Gamma (1+b)}{\Gamma (1+b-\alpha )}t^{b-\alpha },\;\;\mathrm{for}\;b\geq \left[\alpha \right].$$

### The Shifted Legendre Polynomials (LP)

These polynomials play a central role in approximation theory. Generally, these polynomials are defined on the domain [0,τ], which is given by
(3)$$\rho_{l}^{\tau }\left(t\right)=\sum_{b=0}^{l}{ℷ}_{\left(l,b\right)}t^{b},$$
where
(4)$${ℷ}_{\left(l,b\right)}=\frac{{\left(-1\right)}^{l+b}\left(l+b\right)!}{\left(l-b\right)!\tau^{b}{\left(b!\right)}^{2}}.$$
These polynomials enjoys a very important property of orthogonality on the domain [0,τ], which is expressed mathematically as
(5)$$\int_{0}^{\tau }\rho_{i}^{\tau }\left(t\right)\rho_{j}^{\tau }\left(t\right)dt=\left\{\begin{array}{cc} \frac{\tau }{2j+1}, & \mathrm{if}\;i=j,\\ 0, & \mathrm{if}\;i\neq j. \end{array}\right.$$
By using Equation (Equation 5), any s(t) can be expressed in terms of LP as
(6)$$\;s\left(t\right)\approx \;\sum_{l=0}^{m}c_{l}\rho_{l}^{\tau }\left(t\right),\;\mathrm{where}\;\;c_{l}=\frac{\left(2l+1\right)}{\tau }\int_{0}^{\tau }s\left(t\right)\rho_{l}^{\tau }\left(t\right)dt.$$

The above equation has an equivalent vector representation given as
(7)$$\;s\left(t\right)\approx C_{M}\Lambda_{M}^{\tau }\left(t\right),$$
where
(8)$$\Lambda_{M}^{\tau }\left(t\right)={\left[\begin{array}{cccccc} \rho_{0}^{\tau }\left(t\right) & \rho_{1}^{\tau }\left(t\right) & \cdots & \rho_{i}^{\tau }\left(t\right) & \cdots & \rho_{m}^{\tau }\left(t\right) \end{array}\right]}^{T}$$
and
(9)$$C_{M}\;=\;\left[\begin{array}{cccccc} c_{0} & c_{1} & \cdots & c_{i} & \cdots & c_{m} \end{array}\right].$$
The following useful constant is important in the derivation of the operational matrices. We only recall the definition of the constant. The detailed derivation of which can be found in [25].

**Lemma** **1.**
*The integral of product of any three LP is a constant, represented by, ${℘}^{\left(i,j,k\right)},$ defined as*

$$\int_{0}^{\tau }\rho_{i}^{\tau }\left(t\right)\rho_{j}^{\tau }\left(t\right)\rho_{k}^{\tau }\left(t\right)dt={℘}^{\left(i,j,k\right)},$$

*where*

$${℘}^{\left(i,j,k\right)}=\sum_{a=0}^{i}\sum_{p=0}^{j}\sum_{r=0}^{k}{ℷ}_{\left(i,a\right)}{ℷ}_{\left(i,p\right)}{ℷ}_{\left(i,r\right)}Y_{\left(a,p,r\right)}.$$

*${ℷ}_{\left(.,.\right)}$ are as defined in (Equation 4) and*

$$Y_{\left(a,p,r\right)}=\frac{\tau^{\left(a+p+r+1\right)}}{\left(a+p+r+1\right)}.$$



The constant defined above is important in the solution of FDEs. We recall one more important result of the Legendre polynomials, which is their application in the study of convergence and developing of error bounds.

**Theorem** **1**([21])**.**
*Let $\prod_{M}$ be the space of M terms Legendre polynomials and let $u\left(t\right)\in C^{m}\left[0,1\right],$ then $u_{m}\left(t\right)$ is in space $\prod_{M}$; then, for the m term approximation,*
$$u\left(t\right)=\sum_{i=0}^{m}c_{i}\rho_{i}^{\tau }\left(t\right),$$
*there exists a constant C such that*
$$c_{k}≃\frac{C}{\lambda_{k}}\left|u^{\left(m\right)}\right|.$$
*and*
$$|u\left(t\right)-\sum_{i=0}^{m}c_{i}\rho_{i}^{\tau }{\left(t\right)|}^{2}\leq \sum_{k=m+1}^{\infty }\lambda_{k}c_{k}^{2},$$
*where*
$$c_{k}=\frac{\lambda_{k}+1}{\tau }\int_{0}^{\tau }u\left(t\right)\rho_{k}^{\tau }\left(t\right)dt,\;\lambda_{k}=k\left(k+1\right).$$
*C is constant and can be chosen in such a way that $u^{\left(2m\right)}$ belongs to $\prod_{M}$, where $u^{\left(m\right)}$ is defined as*

$$u^{\left(m\right)}=Z\left(u^{\left(m-1\right)}\right)=Z^{m}\left(u^{\left(0\right)}\right)$$

*where **Z** is storm livoliel operator of legendre polynomials with $u^{\left(0\right)}=u\left(t\right).$*


In the next section, we first recall some of our previously designed operational matrices and then develop new operational matrices.

## 3. Operational Matrices (OP)

OP acts as a basic block in developing approximation techniques. The purpose of operational matrices is to replace a given derivative term with its matrix notation. The following matrices are important in our further investigation.

**Lemma** **2**([24])**.**
*Let $\Lambda_{M}^{\tau }\left(t\right)$ be the function vector; the α order integration is generalized as*
$$I^{\alpha }\left(\Lambda_{M}^{\tau }\left(t\right)\right)≃H_{M\times M}^{\tau ,\alpha }\Lambda_{M}^{\tau }\left(t\right),$$
*where $H_{M\times M}^{\tau ,\alpha }$ is the OP of integration, defined as*
$$H_{M\times M}^{\tau ,\alpha }\;=\;\left[\begin{array}{c} \Theta_{i,j,\tau } \end{array}\right],$$
*where*
$$\Theta_{i,j,\tau }=\sum_{a=0}^{i}s_{\left(a,j\right)}{ℷ}_{\left(i,a\right)}\frac{\Gamma (a+1)}{\Gamma (a+\alpha +1)}.$$
*where*
$$s_{\left(a,j\right)}=\frac{\left(2j+1\right)}{\tau }\sum_{l=0}^{j}\frac{{\left(-1\right)}^{j+l}\left(j+l\right)!{\left(\tau \right)}^{a+l+\alpha +1}}{\left(\tau^{l}\right)\left(j-l\right){\left(l!\right)}^{2}\left(a+l+\alpha +1\right)}.$$

**Corollary** **1.**
*The error $|E_{M}\left|=\right|I^{\alpha }u\left(t\right)-C_{M}H_{M\times M}^{\tau ,\alpha }\Lambda_{M}^{\tau }\left(t\right)|$ is bounded by the following relation*

$$|E_{M}\left|\leq \right|\sum_{a=m+1}^{\infty }c_{k}\left\{\sum_{i=0}^{m}\Theta_{a,j,\tau }\right\}|.$$



**Proof.** Consider
$$u\left(t\right)=\sum_{a=0}^{\infty }c_{a}\rho_{a}^{\tau }\left(t\right).$$
Then, using the previous result, we get
$$I^{\alpha }u\left(t\right)=\sum_{a=0}^{\infty }c_{a}\sum_{j=0}^{m}\Theta_{a,j,\tau }\rho_{j}^{\tau }\left(t\right).$$
Simplifying the above estimate$$I^{\alpha }u\left(t\right)-\sum_{a=0}^{m}c_{a}\sum_{j=0}^{m}\Theta_{a,j,\tau }\rho_{j}^{\tau }\left(t\right)=\sum_{a=m+1}^{\infty }c_{a}\sum_{j=0}^{m}\Theta_{a,j,\tau }\rho_{j}^{\tau }\left(t\right).$$
In matrix notation
$$I^{\alpha }u\left(t\right)-C_{M}H_{M\times M}^{\tau ,\alpha }\Lambda_{M}^{\tau }\left(t\right)=\sum_{a=m+1}^{\infty }c_{a}\sum_{j=0}^{m}\Theta_{a,j,\tau }\rho_{j}^{\tau }\left(t\right).$$
Using the fact $|\rho_{j}^{\tau }\left(t\right)|\leq 1$ for t∈[0,τ], therefore, we can write
$$|I^{\alpha }u\left(t\right)-C_{M}H_{M\times M}^{\tau ,\alpha }\Lambda_{M}^{\tau }\left(t\right)|\leq |\sum_{a=m+1}^{\infty }c_{a}\sum_{j=0}^{m}\Theta_{a,j,\tau }|.$$ □

**Lemma** **3**([24])**.**
*Let $\Lambda_{M}^{\tau }\left(t\right)$ be the function vector as defined in (Equation 8); then the derivative of order σ of $\Lambda_{M}^{\tau }\left(t\right)$ is generalized as*
$${}^{c}D^{\sigma }\left(\Lambda_{M}^{\tau }\left(t\right)\right)≃G_{M\times M}^{\tau ,\sigma }\Lambda_{M}^{\tau }\left(t\right),$$
*where $G_{M\times M}^{\tau ,\sigma }$ is the operational matrix of derivative of order σ, and $G_{M\times M}^{\tau ,\sigma }$ is defined as*
$$G_{M\times M}^{\tau ,\sigma }\;=\;\left[\begin{array}{c} \Phi_{i,j,\tau } \end{array}\right]$$
*where*
$$\Phi_{i,j,\tau }=\sum_{k=\lceil \sigma \rceil }^{i}s_{\left(k,j\right)}{ℷ}_{\left(i,k\right)}\frac{\Gamma (k+1)}{\Gamma (k-\sigma +1)}$$
*with $\Phi_{i,j,\tau }=0$ if i<⌈σ⌉.*
*Furthermore, ${ℷ}_{\left(i,k\right)}$ is similar as defined in (Equation 4) and*

$$s_{\left(k,j\right)}=\frac{\left(2j+1\right)}{\tau }\sum_{l=0}^{j}\frac{{\left(-1\right)}^{j+l}\left(j+l\right)!{\left(\tau \right)}^{k+l-\sigma +1}}{\left(\tau^{l}\right)\left(j-l\right){\left(l!\right)}^{2}\left(k+l-\sigma +1\right)}.$$



**Corollary** **2.**
*The error $|E_{M}{\left|\;=\;\right|}^{c}D^{\sigma }u\left(t\right)-C_{M}G_{M\times M}^{\tau ,\sigma }\Lambda_{M}^{\tau }\left(t\right)|$ in approximating $D^{\sigma }u\left(t\right)$ with operational matrix of derivative is bounded by the following.*

$$|E_{M}\left|\leq \right|\sum_{k=m+1}^{\infty }u_{k}\left\{\sum_{i=\lceil \sigma \rceil }^{m}\Phi_{i,j,\tau }\right\}|$$



**Proof.** The proof of this corollary is similar as Corollary 1. □

**Lemma** **4**([24])**.**
*Let u(t) and $\phi_{n}\left(t\right)$ be smooth functions that are well-defined on [0,τ]. Then*
$$\phi_{n}\left(t\right){\;}^{c}D^{\sigma }u\left(t\right)=W_{M}B_{\phi_{n}}^{\sigma }\Lambda_{M}^{\tau }\left(t\right).$$
*where $W_{M}$ is the Legendre coefficients vector of u(t) as defined in (Equation 7) and*
$$B_{\phi_{n}}^{\sigma }=G_{M\times M}^{\tau ,\sigma }J_{M\times M}^{\tau ,\phi_{n}}.$$
$$B_{\phi_{n}}^{\sigma }\;=\;\left[\begin{array}{c} \overbrace{\Theta_{\left(r,j\right)}} \end{array}\right]$$
*where*
$$\overbrace{\Theta_{\left(i,j\right)}}=\sum_{l=0}^{m}\Phi_{\left(i,l\right)}\Omega_{\left(l,j\right)}$$
*The matrix $G_{M\times M}^{\tau ,\sigma }$ is the operational matrix of derivative as defined in Lemma 3; the entries of matrix $J_{M\times M}^{\tau ,\phi_{n}}$ are defined by the following relation*
$$\Omega_{\left(l,j\right)}=\frac{2j+1}{\tau }\sum_{i=0}^{m}c_{i}{℘}_{}^{\left(i,l,j\right)}.$$
*and $c_{i}=\int_{0}^{\tau }\phi_{n}\left(t\right)\rho_{i}\left(t\right)dt$.*

**Corollary** **3.**
*The error $|E_{M}\left|\;=\;\right|\phi_{n}{\left(t\right)}^{c}D^{\sigma }u\left(t\right)-C_{M}B_{\phi_{n}}^{\sigma }\Lambda_{M}^{\tau }\left(t\right)|$ in approximating $\phi_{n}{\left(t\right)}^{c}D^{\sigma }u\left(t\right)$ with operational matrix of fractional derivative with variable coefficient is bounded by the following.*

$$|E_{M}\left|\leq \right|\sum_{k=m+1}^{\infty }c_{k}\sum_{j=0}^{m}\overbrace{\Theta_{\left(k,j\right)}}|.$$



**Proof.** Consider
$$u\left(t\right)=\sum_{k=0}^{\infty }c_{k}\rho_{k}^{\tau }\left(t\right).$$
Then, using the relation above, we get$$\phi_{n}{\left(t\right)}^{c}D^{\sigma }u\left(t\right)=\sum_{k=0}^{\infty }c_{k}\sum_{j=0}^{m}\overbrace{\Theta_{\left(k,j\right)}}\rho_{j}^{\tau }\left(t\right)$$
Truncating the sum and writing in modified form we get
$$\phi_{n}{\left(t\right)}^{c}D^{\sigma }u\left(t\right)-\sum_{k=0}^{m}c_{k}\sum_{j=0}^{m}\overbrace{\Theta_{\left(k,j\right)}}\rho_{j}^{\tau }\left(t\right)=\sum_{k=m+1}^{\infty }c_{k}\sum_{j=0}^{m}\overbrace{\Theta_{\left(k,j\right)}}\rho_{j}^{\tau }\left(t\right).$$
We can also write it in matrix form as$$\phi_{n}{\left(t\right)}^{c}D^{\sigma }u\left(t\right)-C_{M}B_{\phi_{n}}^{\sigma }\Lambda_{M}^{\tau }\left(t\right)=\sum_{k=m+1}^{\infty }c_{k}\sum_{j=0}^{m}\overbrace{\Theta_{\left(k,j\right)}}\rho_{j}^{\tau }\left(t\right)$$
Using the fact $\rho_{j}^{\tau }\left(t\right)\leq 1$ for t∈[0,τ], therefore, we can write
$$|\phi_{n}{\left(t\right)}^{c}D^{\sigma }u\left(t\right)-C_{M}B_{\phi_{n}}^{\sigma }\Lambda_{M}^{\tau }\left(t\right)|\;\leq \;|\sum_{k=m+1}^{\infty }c_{k}\sum_{j=0}^{m}\overbrace{\Theta_{\left(k,j\right)}}|.$$
Furthermore, hence the proof is complete. □

The above matrices have been successfully applied to solve fractional-order differential equations (FDEs) under the effect of initial conditions. However these matrices do not have the ability to solve FDEs with integral types of boundary conditions. Therefore, the invention of new matrices that can easily handle integral types of boundary conditions are of basic importance. In the forthcoming discussion, we will design two new operational matrices having the ability to deal with integral type boundary conditions.

**Lemma** **5.**
*Let s(t) be a known function well defined on [0,τ] and $\phi_{c}^{n}\left(t\right)=ct^{n}$ be a polynomial then, for a function vector $\Lambda_{M}^{\tau }\left(t\right)$ as defined in (Equation 8), the following result holds*

$$\phi_{c}^{n}\left(t\right)\int_{0}^{\tau }s\left(t\right)\Lambda_{M}^{\tau }\left(t\right)dt=Q_{M\times M}^{c,n,s(t)}\Lambda_{M}^{\tau }\left(t\right),$$

*where the matrix $Q_{M\times M}^{c,n,s(t)}$ is given as*

$$Q_{M\times M}^{c,n,s(t)}=\left(\begin{array}{cccc} \Omega_{\left(0,0\right)} & \Omega_{\left(0,1\right)} & \cdots & \Omega_{\left(0,m\right)}\\ \Omega_{\left(1,0\right)} & \Omega_{\left(1,1\right)} & \cdots & \Omega_{\left(1,m\right)}\\ \vdots & \vdots & \ddots & \vdots \\ \Omega_{\left(m,0\right)} & \Omega_{\left(m,1\right)} & \cdots & \Omega_{\left(m,m\right)} \end{array}\right).$$

*where*

$$\Omega_{\left(i,j\right)}=\sum_{r=0}^{j}\frac{\tau cd_{i}\tau^{r+n+1}{ℷ}_{\left(j,r\right)}}{\left(2i+1)(r+n+1\right)},$$

*$d_{i}$ is the Legendre coefficients of the function s(t) and ${ℷ}_{\left(j,r\right)}$ is as defined in (Equation 4).*


**Proof.** Let s(t) be approximated with Legendre polynomials, as
$$s\left(t\right)=\sum_{l=0}^{m}d_{l}\phi_{c}^{n}\left(t\right).$$
We can write the $i^{th}$ term of $\phi_{c}^{n}\left(t\right)\int_{0}^{\tau }s\left(t\right)\Lambda_{M}^{\tau }\left(t\right)dt$ as
$$\begin{array}{cc} ct^{n}\int_{0}^{\tau }\sum_{l=0}^{m}d_{l}\rho_{l}^{\tau }\left(t\right)\rho_{i}^{\tau }\left(t\right)dt & =ct^{n}\sum_{l=0}^{m}d_{l}\int_{0}^{\tau }\rho_{l}^{\tau }\left(t\right)\rho_{i}^{\tau }\left(t\right)dt,\\ \; & =\frac{\tau cd_{i}t^{n}}{2i+1}. \end{array}$$
The polynomial $\frac{\tau cd_{i}t^{n}}{2i+1}$ can be expressed as a series of Legendre polynomials as
$$\frac{\tau cd_{i}t^{n}}{2i+1}=\sum_{j=0}^{m}\Omega_{\left(i,j\right)}\rho_{j}\left(t\right).$$
where the constant $\Omega_{\left(i,j\right)}$ is given by
$$\Omega_{\left(i,j\right)}=\frac{\tau cd_{i}}{2i+1}\int_{0}^{\tau }\rho_{j}^{\tau }\left(t\right)t^{n}dt.$$
Using the definition of $\rho_{j}^{\tau }\left(t\right)$ and after simplification, we can write
$$\begin{array}{cc} \Omega_{\left(i,j\right)} & =\frac{\tau cd_{i}}{2i+1}\sum_{r=0}^{j}{ℷ}_{\left(j,r\right)}\int_{0}^{\tau }t^{r+n}dt,\\ \; & =\sum_{r=0}^{j}\frac{\tau cd_{i}\tau^{r+n+1}{ℷ}_{\left(j,r\right)}}{\left(2i+1)(r+n+1\right)}. \end{array}$$
Simulating the result for i=0:M and j=0:M completes the proof of the Lemma. □

**Lemma** **6.**
*Let $\phi_{c}^{n}\left(n\right)=ct^{n}$ be a polynomial then for a function vector $\Lambda_{M}^{\tau }\left(t\right)$ as defined in (Equation 8); the following holds*

$$\phi_{c}^{n}\left(t\right)\Lambda_{M}^{\tau }\left(\tau \right)=R_{M\times M}^{c,n,\tau }\Lambda_{M}^{\tau }\left(t\right)$$

*The matrix $R_{M\times M}^{c,n,\tau }$ is defined as*

$$R_{M\times M}^{c,n,\tau }=\left(\begin{array}{cccc} {ℑ}_{\left(0,0\right)} & {ℑ}_{\left(0,1\right)} & \cdots & {ℑ}_{\left(0,m\right)}\\ {ℑ}_{\left(1,0\right)} & {ℑ}_{\left(1,1\right)} & \cdots & {ℑ}_{\left(1,m\right)}\\ \vdots & \vdots & \ddots & \vdots \\ {ℑ}_{\left(m,0\right)} & {ℑ}_{\left(m,1\right)} & \cdots & {ℑ}_{\left(m,m\right)} \end{array}\right).$$

*Furthermore, the entries are defined as*

$${ℑ}_{\left(i,j\right)}=\sum_{k=0}^{i}\sum_{l=0}^{j}\frac{c\tau^{k+n+l+1}{ℷ}_{\left(i,k\right)}{ℷ}_{\left(j,l\right)}}{n+l+1}.$$



**Proof.** The general term of the equality can be written as
$$\phi_{c}^{n}\left(t\right)\rho_{i}^{\tau }\left(\tau \right)=\sum_{k=0}^{i}{ℷ}_{\left(i,k\right)}\tau^{k}ct^{n}.$$
It can be approximated with Legendre polynomials
$$\sum_{k=0}^{i}{ℷ}_{\left(i,k\right)}\tau^{k}ct^{n}=\sum_{j=0}^{m}{ℑ}_{\left(i,j\right)}\rho_{j}^{\tau }\left(t\right),$$
where
$${ℑ}_{\left(i,j\right)}=\int_{0}^{\tau }\sum_{k=0}^{i}{ℷ}_{\left(i,k\right)}\tau^{k}ct^{n}\rho_{j}^{\tau }\left(t\right)dt.$$
Using the definition of Legendre polynomials we can write
$${ℑ}_{\left(i,j\right)}=\sum_{k=0}^{i}\sum_{l=0}^{j}\frac{c\tau^{k+n+l+1}{ℷ}_{\left(i,k\right)}{ℷ}_{\left(j,l\right)}}{n+l+1}.$$which completes the proof of the lemma. □

## 4. Application of Operational Matrices

The operational matrix method for the solution of fractional differential equations is, in fact, a spectral method. The main aim of the spectral method is to convert a typical differential equation to system of easily solvable algebraic equations, which can be solved to obtain the solution in the series form of some orthogonal polynomials. The application of these methods to nonlinear differential equations results in a nonlinear system of algebraic equations, which can be solved using some iterative algorithms (the Newton–Raphson method is a frequently used method), see for example [46,47,48,49,50,51,52,53].

In this paper, we implement a different approach. We first use the Taylor series method to linearize the nonlinear part *f* and *g* to convert the nonlinear fractional differential equation into a recurrence relation of linear fractional differential equations with variable coefficients.

### 4.1. Linear FDEs with Variable Coefficients

We first consider the following coupled system of linear fractional differential equations with variable coefficients
(10)$$\begin{array}{cc} \; & {}^{c}D^{\sigma_{1}}u\left(t\right)=c_{0}\left(t\right)u+c_{1}\left(t\right)u^{\left(\gamma_{1}\right)}+c_{2}\left(t\right)v+c_{3}\left(t\right)v^{\left(\gamma_{1}\right)}+h\left(t\right),\\ \; & {}^{c}D^{\sigma_{2}}v\left(t\right)=d_{0}\left(t\right)u+d_{1}\left(t\right)u^{\left(\gamma_{1}\right)}+d_{2}\left(t\right)v+d_{3}\left(t\right)v^{\left(\gamma_{1}\right)}+k\left(t\right),\\ \; & u\left(0\right)=u_{0},\;u\left(\tau \right)=m_{1}\int_{0}^{\tau }s\left(t\right)u\left(t\right)dt,\\ \; & v\left(0\right)=v_{0},\;v\left(\tau \right)=m_{2}\int_{0}^{\tau }r\left(t\right)v\left(t\right)dt, \end{array}$$
where $m_{1}$ and $m_{2}$ are some real constants. $0<\gamma_{1},\gamma_{2}\leq 1,$
$1<\sigma_{1},\sigma_{2}\leq 2,$ t∈[0,τ] and $c_{i}\left(t\right),d_{i}\left(t\right),s\left(t\right),r\left(t\right),h\left(t\right)$ and k(t) are bounded, continuous, and well-defined functions on the domain [0,τ].

Assume the solution of (Equation 10) in terms of shifted Legendre polynomials, such that the following hold
(11)$${}^{c}D^{\sigma_{1}}u\left(t\right)=K_{M}\Lambda_{M}^{\tau }\left(t\right),{\;}^{c}D^{\sigma_{2}}v\left(t\right)=L_{M}\Lambda_{M}^{\tau }\left(t\right).$$
Applying fractional integral of order $\sigma_{1}$ on the first equation of (Equation 11) and making use of Lemma 2, we can write
(12)$$u\left(t\right)=K_{M}H_{M\times M}^{\tau ,\sigma_{1}}\Lambda_{M}^{\tau }\left(t\right)+e_{0}+e_{1}t.$$
By the application of initial conditions, we get $e_{0}=u_{0},$ and to get the value of $e_{1}$, we use the integral-type boundary conditions, and after simplification it follows that
$$\begin{array}{cc} K_{M} & H_{M\times M}^{\tau ,\sigma_{1}}\Lambda_{M}^{\tau }\left(\tau \right)+u_{0}+e_{1}\tau =\\ & m_{1}K_{M}H_{M\times M}^{\tau ,\sigma_{1}}\int_{0}^{\tau }s\left(t\right)\Lambda_{M}^{\tau }\left(t\right)dt\\ & +m_{1}\int_{0}^{\tau }s\left(t\right)u_{0}dt+m_{1}e_{1}\int_{0}^{\tau }s\left(t\right)tdt, \end{array}$$
$$\begin{array}{cc} \; & e_{1}=\frac{1}{s_{1}}(K_{M}H_{M\times M}^{\tau ,\sigma_{1}}\Lambda_{M}^{\tau }\left(\tau \right)\\ & -m_{1}K_{M}H_{M\times M}^{\tau ,\sigma_{1}}\int_{0}^{\tau }s\left(t\right)\Lambda_{M}^{\tau }\left(t\right)dt-s_{0}), \end{array}$$
where $s_{1}=\left(m_{1}\int_{0}^{\tau }s\left(t\right)tdt-\tau \right)$ and $s_{0}=m_{1}\int_{0}^{\tau }s\left(t\right)u_{0}dt-u_{0}.$

Now using the values of $e_{0}$ and $e_{1}$ in (Equation 12), we can write u(t) as
(13)$$\begin{array}{cc} \; & u\left(t\right)=K_{M}H_{M\times M}^{\tau ,\sigma_{1}}\Lambda_{M}^{\tau }\left(t\right)+u_{0}+\\ & \frac{t}{s_{1}}\left(K_{M}H_{M\times M}^{\tau ,\sigma_{1}}\Lambda_{M}^{\tau }\left(\tau \right)-m_{1}K_{M}H_{M\times M}^{\tau ,\sigma_{1}}\int_{0}^{\tau }s\left(t\right)\Lambda_{M}^{\tau }\left(t\right)dt-s_{0}\right), \end{array}$$
In view of Lemma 5 and Lemma 6, we can write Equation (Equation 13) as
$$\begin{array}{cc} u(t)= & K_{M}H_{M\times M}^{\tau ,\sigma_{1}}\Lambda_{M}^{\tau }\left(t\right)+K_{M}H_{M\times M}^{\tau ,\sigma_{1}}R_{M\times M}^{\frac{1}{s_{1}},1,\tau }\Lambda_{M}^{\tau }\left(t\right)\\ & -K_{M}H_{M\times M}^{\tau ,\sigma_{1}}Q_{M\times M}^{\frac{m_{1}}{s_{1}},1,s\left(t\right)}\Lambda_{M}^{\tau }\left(t\right)+{F_{1}}_{M}\Lambda_{M}^{\tau }\left(t\right). \end{array}$$
where $u_{0}-\frac{s_{0}t}{s_{1}}=F_{1}\Lambda_{M}^{\tau }\left(t\right).$ For simplicity of notation, we can write the above equations as
(14)$$u\left(t\right)=K_{M}{E^{\left(1\right)}}_{M\times M}\Lambda_{M}^{\tau }\left(t\right)+{F_{1}}_{M}\Lambda_{M}^{\tau }\left(t\right).$$
where
$${E^{\left(1\right)}}_{M\times M}=H_{M\times M}^{\tau ,\sigma_{1}}\left(I_{M\times M}+R_{M\times M}^{\frac{1}{s_{1}},1,\tau }-Q_{M\times M}^{\frac{m_{1}}{s_{1}},1,s\left(t\right)}\right)$$
Repeating the same procedure for v(t), we can get analogous representation as
(15)$$v\left(t\right)=L_{M}{E^{\left(2\right)}}_{M\times M}\Lambda_{M}^{\tau }\left(t\right)+{F_{2}}_{M}\Lambda_{M}^{\tau }\left(t\right).$$
where
$$E_{M\times M}^{\left(2\right)}=H_{M\times M}^{\tau ,\sigma_{2}}\left(I_{M\times M}+R_{M\times M}^{\frac{1}{r_{1}},1,\tau }-Q_{M\times M}^{\frac{m_{2}}{r_{1}},1,r\left(t\right)}\right)$$
Now, in view of (Equation 15), (Equation 14), (Equation 11), and Lemma 4, the equivalent matrix form for system (Equation 10) is given as
(16)$$\begin{array}{cc} K_{M}\Lambda_{M}^{\tau }\left(t\right)= & K_{M}E_{M\times M}^{\left(1\right)}\left(B_{M\times M}^{c_{0},0}+B_{M\times M}^{c_{1},\gamma_{1}}\right)\Lambda_{M}^{\tau }\left(t\right)+\\ & {F_{1}}_{M}\left(B_{M\times M}^{c_{0},0}+B_{M\times M}^{c_{1},\gamma_{1}}\right)\Lambda_{M}^{\tau }\left(t\right)+\\ \; & L_{M}E_{M\times M}^{\left(2\right)}\left(B_{M\times M}^{c_{2},0}+B_{M\times M}^{c_{3},\gamma_{2}}\right)\Lambda_{M}^{\tau }\left(t\right)+\\ & {F_{2}}_{M}\left(B_{M\times M}^{c_{2},0}+B_{M\times M}^{c_{3},\gamma_{2}}\right)\Lambda_{M}^{\tau }\left(t\right)+{Z_{1}}_{M}\Lambda_{M}^{\tau }\left(t\right),\\ L_{M}\Lambda_{M}^{\tau }\left(t\right)= & K_{M}E_{M\times M}^{\left(1\right)}\left(B_{M\times M}^{d_{0},0}+B_{M\times M}^{d_{1},\gamma_{1}}\right)\Lambda_{M}^{\tau }\left(t\right)+\\ & {F_{1}}_{M}\left(B_{M\times M}^{d_{0},0}+B_{M\times M}^{d_{1},\gamma_{1}}\right)\Lambda_{M}^{\tau }\left(t\right)+\\ \; & L_{M}E_{M\times M}^{\left(2\right)}\left(B_{M\times M}^{d_{2},0}+B_{M\times M}^{d_{3},\gamma_{2}}\right)\Lambda_{M}^{\tau }\left(t\right)+\\ & {F_{2}}_{M}\left(B_{M\times M}^{d_{2},0}+B_{M\times M}^{d_{3},\gamma_{2}}\right)\Lambda_{M}^{\tau }\left(t\right)+{Z_{2}}_{M}\Lambda_{M}^{\tau }\left(t\right) \end{array}$$
Canceling out the common term and after simplification of notation, we can write
(17)$$\begin{array}{cc} K_{M}= & K_{M}E_{M\times M}^{\left(1\right)}\left(B_{M\times M}^{c_{0},0}+B_{M\times M}^{c_{1},\gamma_{1}}\right)+\\ & L_{M}E_{M\times M}^{\left(2\right)}\left(B_{M\times M}^{c_{2},0}+B_{M\times M}^{c_{3},\gamma_{2}}\right)+{Y_{1}}_{M}\\ L_{M}= & K_{M}E_{M\times M}^{\left(1\right)}\left(B_{M\times M}^{d_{0},0}+B_{M\times M}^{d_{1},\gamma_{1}}\right)+\\ & L_{M}E_{M\times M}^{\left(2\right)}\left(B_{M\times M}^{d_{2},0}+B_{M\times M}^{d_{3},\gamma_{2}}\right)+{Y_{2}}_{M}, \end{array}$$
where
$${Y_{1}}_{M}={F_{2}}_{M}\left(B_{M\times M}^{c_{2},0}+B_{M\times M}^{c_{3},\gamma_{2}}\right)+$$
$${F_{1}}_{M}\left(B_{M\times M}^{c_{0},0}+B_{M\times M}^{c_{1},\gamma_{1}}\right)+{Z_{1}}_{M},$$
and
$${Y_{2}}_{M}={F_{2}}_{M}\left(B_{M\times M}^{d_{2},0}+B_{M\times M}^{d_{3},\gamma_{2}}\right)+$$
$${F_{1}}_{M}\left(B_{M\times M}^{d_{0},0}+B_{M\times M}^{d_{1},\gamma_{1}}\right)+{Z_{2}}_{M}.$$
The above equations can also be written as
(18)$$\begin{array}{cc} \; & \left[K_{M}\;L_{M}\right]\;=\;\left[K_{M}\;L_{M}\right]\\ & \left[\begin{array}{cc} E^{\left(1\right)}\left(B^{c_{0},0}+B^{c_{1},\gamma_{1}}\right) & E^{\left(1\right)}\left(B^{d_{0},0}+B^{d_{1},\gamma_{1}}\right)\\ E^{\left(2\right)}\left(B^{c_{2},0}+B^{c_{3},\gamma_{2}}\right) & E^{\left(2\right)}\left(B^{d_{2},0}+B^{d_{3},\gamma_{2}}\right) \end{array}\right]\\ & +\left[\begin{array}{cc} {Y_{1}}_{M} & {Y_{2}}_{M} \end{array}\right] \end{array}$$

We see that (Equation 18) is a linear system of matrix equations. By solving (Equation 17), we will get the required coefficients vector $K_{M}$ and $L_{M}$, which can be used in (Equation 14) and (Equation 15) to get approximation to the solution of the main problem.

### 4.2. Nonlinear FDEs

Nonlinear FDEs cannot be directly solved using the OP method; however, combining it with the quasilinearization method makes it easy to recursively solve nonlinear FDEs. The procedure of this technique is as given as follows.
Approximate the initial solution, the solution of the linear part, by the method presented in previous section and name it $u_{0}\left(t\right)$ and $v_{0}\left(t\right).$Linearize the nonlinear part at $u_{0}\left(t\right)$ and $v_{0}\left(t\right)$. This will convert the system of nonlinear FDEs into a system of linear FDEs that is easily solvable with the method devolved. Solve it and name the solution as $u_{1}\left(t\right)$ and $v_{1}\left(t\right)$.Repeat step 1.

Consider the following nonlinear FDEs.
(19)$$\begin{array}{cc} \; & {}^{c}D^{\sigma_{1}}u\left(t\right)=f\left(u,v,u^{\left(\gamma_{1}\right)},v^{\left(\gamma_{2}\right)}\right),\\ \; & {}^{c}D^{\sigma_{2}}v\left(t\right)=g\left(u,v,u^{\left(\gamma_{1}\right)},v^{\left(\gamma_{2}\right)}\right),\\ \; & u\left(0\right)=a_{0},\;u\left(\tau \right)=\int_{0}^{\omega_{1}}s\left(t\right)u\left(t\right)dt,\;0<\omega_{1}\leq \tau ,\\ \; & v\left(0\right)=b_{0},\;v\left(\tau \right)=\int_{0}^{\omega_{2}}r\left(t\right)v\left(t\right)dt,\;0<\omega_{2}\leq \tau . \end{array}$$
separating the linear and nonlinear parts, we get
(20)$$\begin{array}{cc} \; & {}^{c}D^{\sigma_{1}}u\left(t\right)=L_{1}\left(u,v,u^{\left(\gamma_{1}\right)},v^{\left(\gamma_{2}\right)}\right)+N_{1}\left(u,v,u^{\left(\gamma_{1}\right)},v^{\left(\gamma_{2}\right)}\right),\\ \; & {}^{c}D^{\sigma_{2}}v\left(t\right)=L_{2}\left(u,v,u^{\left(\gamma_{1}\right)},v^{\left(\gamma_{2}\right)}\right)+N_{2}\left(u,v,u^{\left(\gamma_{1}\right)},v^{\left(\gamma_{2}\right)}\right), \end{array}$$
First solve the linear part:
(21)$$\begin{array}{cc} \; & {}^{c}D^{\sigma_{1}}u\left(t\right)=L_{1}\left(u,v,u^{\left(\gamma_{1}\right)},v^{\left(\gamma_{2}\right)}\right),\\ \; & {}^{c}D^{\sigma_{2}}v\left(t\right)=L_{2}\left(u,v,u^{\left(\gamma_{1}\right)},v^{\left(\gamma_{2}\right)}\right), \end{array}$$
Its solution is named $u_{0}\left(t\right)$ and $v_{0}\left(t\right)$. The next step is to linearize the nonlinear part.
(22)$$\begin{array}{cc} {}^{c}D^{\sigma_{1}}u_{1}\left(t\right) & =L_{1}\left(u_{1},v_{1},u_{1}^{\left(\gamma_{1}\right)},v_{1}^{\left(\gamma_{2}\right)}\right)+N_{1}\left(u_{0},v_{0},u_{0}^{\left(\gamma_{1}\right)},v_{0}^{\left(\gamma_{2}\right)}\right)\\ & +\left(u_{1}-u_{0}\right)\frac{\partial N_{1}}{\partial u_{0}}+\left(v_{1}-v_{0}\right)\frac{\partial N_{1}}{\partial v_{0}}+\\ \; & \left(u_{1}^{\left(\gamma_{1}\right)}-u_{0}^{\left(\gamma_{1}\right)}\right)\frac{\partial N_{1}}{\partial u_{0}^{\left(\gamma_{1}\right)}}+\left(v_{1}^{\left(\gamma_{2}\right)}-v_{0}^{\left(\gamma_{2}\right)}\right)\frac{\partial N_{1}}{\partial v_{0}^{\left(\gamma_{2}\right)}},\\ {}^{c}D^{\sigma_{2}}v_{1}\left(t\right) & =L_{2}\left(u_{1},v_{1},u_{1}^{\left(\gamma_{1}\right)},v_{1}^{\left(\gamma_{2}\right)}\right)+N_{2}\left(u_{0},v_{0},u_{0}^{\left(\gamma_{1}\right)},v_{0}^{\left(\gamma_{2}\right)}\right)\\ & +\left(u_{1}-u_{0}\right)\frac{\partial N_{2}}{\partial u_{0}}+\left(v_{1}-v_{0}\right)\frac{\partial N_{2}}{\partial v_{0}}+\\ \; & \left(u_{1}^{\left(\gamma_{1}\right)}-u_{0}^{\left(\gamma_{1}\right)}\right)\frac{\partial N_{2}}{\partial u_{0}^{\left(\gamma_{1}\right)}}+\left(v_{1}^{\left(\gamma_{2}\right)}-v_{0}^{\left(\gamma_{2}\right)}\right)\frac{\partial N_{2}}{\partial v_{0}^{\left(\gamma_{2}\right)}}. \end{array}$$
We get a system of FDEs with variable coefficients. The whole process can be expressed as
(23)$$\begin{array}{cc} {}^{c}D^{\sigma_{1}}u_{r+1}\left(t\right) & =L_{1}\left(u_{r+1},v_{r+1},u_{r+1}^{\left(\gamma_{1}\right)},v_{r+1}^{\left(\gamma_{2}\right)}\right)+\\ & N_{1}\left(u_{r},v_{r},u_{r}^{\left(\gamma_{1}\right)},v_{r}^{\left(\gamma_{2}\right)}\right)\\ & +\left(u_{r+1}-u_{r}\right)\frac{\partial N_{1}}{\partial u_{r}}+\left(v_{r+1}-v_{r}\right)\frac{\partial N_{1}}{\partial v_{r}}+\\ \; & \left(u_{r+1}^{\left(\gamma_{1}\right)}-u_{r}^{\left(\gamma_{1}\right)}\right)\frac{\partial N_{1}}{\partial u_{r}^{\left(\gamma_{1}\right)}}+\left(v_{r+1}^{\left(\gamma_{2}\right)}-v_{r}^{\left(\gamma_{2}\right)}\right)\frac{\partial N_{1}}{\partial v_{r}^{\left(\gamma_{2}\right)}}\\ {}^{c}D^{\sigma_{2}}v_{r+1}\left(t\right)\; & =L_{2}\left(u_{r+1},v_{r+1},u_{r+1}^{\left(\gamma_{1}\right)},v_{r+1}^{\left(\gamma_{2}\right)}\right)+\\ & N_{2}\left(u_{r},v_{r},u_{r}^{\left(\gamma_{1}\right)},v_{r}^{\left(\gamma_{2}\right)}\right)+\left(u_{r+1}-u_{r}\right)\frac{\partial N_{2}}{\partial u_{r}}\\ & +\left(v_{r+1}-v_{r}\right)\frac{\partial N_{2}}{\partial v_{r}}+\left(u_{r+1}^{\left(\gamma_{1}\right)}-u_{r}^{\left(\gamma_{1}\right)}\right)\frac{\partial N_{2}}{\partial u_{r}^{\left(\gamma_{1}\right)}}\\ & +\left(v_{r+1}^{\left(\gamma_{2}\right)}-v_{r}^{\left(\gamma_{2}\right)}\right)\frac{\partial N_{2}}{\partial v_{r}^{\left(\gamma_{2}\right)}} \end{array}$$
The boundary conditions can be written as
(24)$$\begin{array}{cc} \; & u_{r+1}\left(0\right)=a_{0},u_{r+1}\left(\tau \right)=\int_{0}^{\omega_{1}}s\left(t\right)u_{r+1}\left(t\right)dt,0<\omega_{1}\leq \tau ,\\ \; & v_{r+1}\left(0\right)=b_{0},v_{r+1}\left(\tau \right)=\int_{0}^{\omega_{2}}r\left(t\right)v_{r+1}\left(t\right)dt,0<\omega_{2}\leq \tau . \end{array}$$

It can be easily noted that (Equation 23) is fractional differential equation with variable coefficients.

## 5. Error Bound of the Approximate Solution and Convergence

In this section, we calculate a upper bound for error of approximation of solution with the proposed method.

### 5.1. Error Bound for Single Differential Equation

Consider the following fractional differential equation.
(25)$${}^{c}D^{\sigma }u\left(t\right)=c_{0}\left(t\right)u\left(t\right)+c_{1}\left(t\right)u{\left(t\right)}^{\left(\gamma \right)}+h\left(t\right),$$
subject to the following initial and boundary conditions
$$u\left(0\right)=u_{0},\;u\left(\tau \right)=m_{1}\int_{0}^{\tau }s\left(t\right)u\left(t\right)dt.$$
Our aim is to derive an upper bound for the proposed method. We have to calculate $R_{M}$ defined as
(26)$$R_{M}={|}^{c}D^{\sigma }u\left(t\right)-K_{M}G_{M\times M}^{\tau ,\sigma }\Lambda_{M}^{\tau }\left(t\right)|.$$

The solution of the above problem can be written in terms of shifted Legendre series such that
(27)$${}^{c}D^{\sigma }u\left(t\right)=\sum_{k=0}^{\infty }u_{k}\rho_{k}^{\tau }\left(t\right)=K_{M}\Lambda_{M}^{\tau }\left(t\right)+\sum_{k=m+1}^{\infty }u_{k}\rho_{k}^{\tau }\left(t\right).$$
Applying a fractional integral of order σ, using an operational matrix of integration and using Corollary (1), we can write
(28)$$u\left(t\right)-c_{0}-c_{1}t=K_{M}H_{M\times M}^{\tau ,\sigma }\Lambda_{M}^{\tau }\left(t\right)+\sum_{k=m+1}^{\infty }c_{k}\sum_{i=0}^{m}\Theta_{i,k,\tau }\rho_{i}^{\tau }\left(t\right).$$
Which can be simplified as
(29)$$u\left(t\right)=K_{M}H_{M\times M}^{\tau ,\sigma }\Lambda_{M}^{\tau }\left(t\right)+c_{0}+c_{1}t+\sum_{k=m+1}^{\infty }u_{k}\sum_{i=0}^{m}\Theta_{i,k,\tau }\rho_{i}^{\tau }\left(t\right),$$
We know from (Equation 14) and the integral type boundary conditions that we can conclude,
$$u\left(t\right)=K_{M}E_{M\times M}\Lambda_{M}^{\tau }\left(t\right)+{F_{1}}_{M}\Lambda_{M}^{\tau }\left(t\right)+\sum_{k=m+1}^{\infty }u_{k}\sum_{i=0}^{m}\Theta_{i,k,\tau }\rho_{i}^{\tau }\left(t\right),$$
Assume that $\hat{X}=K_{M}E_{M\times M}+F_{1}.$ Using Lemma (3) and Corollary (2), we can write
(30)$$c_{l}\left(t\right)u{\left(t\right)}^{\left(\gamma \right)}={\hat{X}}_{M}^{T}B_{c_{l}}^{\gamma }\Lambda_{M}^{\tau }\left(t\right)+\sum_{k=m+1}^{\infty }u_{k}\sum_{i=0}^{m}\sum_{i^{\prime }=0}^{m}\Theta_{i,k,\tau }\overbrace{\Theta_{k,i^{\prime },\tau }^{\left(l\right)}}\rho_{i^{\prime }}^{\tau }\left(t\right).$$
Approximating $h\left(t\right)=F_{2}\Lambda_{M}^{\tau }\left(t\right)+\sum_{k^{\prime }=m+1}^{\infty }f_{k^{\prime }}\rho_{k^{\prime }}^{\tau }\left(t\right)$ and using (Equation 30) in (Equation 25) we get
$$\begin{array}{cc} \; & K_{M}\Lambda_{M}^{\tau }\left(t\right)-{\hat{X}}_{M}^{T}B_{c_{0}}^{0}\Lambda_{M}^{\tau }\left(t\right)-{\hat{X}}_{M}^{T}B_{c_{1}}^{\gamma }\Lambda_{M}^{\tau }\left(t\right)-F_{2}\Lambda_{M}^{\tau }\left(t\right)=R_{M}\left(t\right). \end{array}$$
where $R_{M}\left(t\right)$ is defined by relation
$$\begin{array}{cc} R_{M}\left(t\right)= & \sum_{k=m+1}^{\infty }u_{k}\rho_{k}^{\tau }\left(t\right)+\sum_{k=m+1}^{\infty }u_{k}\sum_{i=0}^{m}\sum_{i^{\prime }=0}^{m}\Theta_{i,k,\tau }\overbrace{\Theta_{k,i^{\prime },\tau }^{\left(0\right)}}\rho_{i^{\prime }}^{\tau }\left(t\right)+\\ & \sum_{k=m+1}^{\infty }u_{k}\sum_{i=0}^{m}\sum_{i^{\prime }=0}^{m}\Theta_{i,k,\tau }\overbrace{\Theta_{k,i^{\prime },\tau }^{\left(1\right)}}\rho_{i^{\prime }}^{\tau }\left(t\right)+\sum_{k^{\prime }=m+1}^{\infty }f_{k^{\prime }}\rho_{k^{\prime }}^{\tau }\left(t\right), \end{array}$$
$$\begin{array}{cc} R_{M}\left(t\right)= & \sum_{k=m+1}^{\infty }u_{k}[\rho_{k}^{\tau }\left(t\right)+\sum_{i=0}^{m}\sum_{i^{\prime }=0}^{m}\Theta_{i,k,\tau }\overbrace{\Theta_{k,i^{\prime },\tau }^{\left(0\right)}}\rho_{i^{\prime }}^{\tau }\left(t\right)+\\ & \sum_{i=0}^{m}\sum_{i^{\prime }=0}^{m}\Theta_{i,k,\tau }\overbrace{\Theta_{k,i^{\prime },\tau }^{\left(1\right)}}\rho_{i^{\prime }}^{\tau }\left(t\right)]+\sum_{k^{\prime }=m+1}^{\infty }f_{k^{\prime }}\rho_{k^{\prime }}^{\tau }\left(t\right). \end{array}$$
Using the bounded property of Legendre polynomail, it follows that
(31)$$\begin{array}{cc} |R_{M}\left(t\right)|\leq & \sum_{k=m+1}^{\infty }|u_{k}\left|\right|\left[1+\sum_{i=0}^{m}\sum_{i^{\prime }=0}^{m}\Theta_{i,k,\tau }\overbrace{\Theta_{k,i^{\prime },\tau }^{\left(0\right)}}+\sum_{i=0}^{m}\sum_{i^{\prime }=0}^{m}\Theta_{i,k,\tau }\overbrace{\Theta_{k,i^{\prime },\tau }^{\left(1\right)}}\right]|+\sum_{k^{\prime }=m+1}^{\infty }\left|f_{k^{\prime }}\right|. \end{array}$$

In view of Theorem (1), it is evident that if the function u(t) and f(t) are sufficiently smooth functions, then the sequence that defines their coefficient is convergent to zero. Hence, we conclude that as m→∞ the coefficients $u_{m}\to 0$ and $f_{m}\to 0$. Hence it can be easily observed that the error $|R_{M}\left(t\right)|\to 0.$ Equation (Equation 31) also establishes an upper bound of the error between the exact and approximate solution.

### 5.2. Error Bound for Coupled System of Fractional Differential Equations

Consider the following system of FDEs.
(32)$$\begin{array}{cc} \; & {}^{c}D^{\sigma }u\left(t\right)=c_{0}\left(t\right)u\left(t\right)+c_{1}\left(t\right)v\left(t\right)+c_{2}\left(t\right)u{\left(t\right)}^{\left(\gamma \right)}+c_{3}\left(t\right)v{\left(t\right)}^{\left(\gamma \right)}+h\left(t\right),\\ \; & {}^{c}D^{\sigma }v\left(t\right)=d_{0}\left(t\right)u\left(t\right)+d_{1}\left(t\right)v\left(t\right)+d_{2}\left(t\right)u{\left(t\right)}^{\left(\gamma \right)}+d_{3}\left(t\right)v{\left(t\right)}^{\left(\gamma \right)}+g\left(t\right), \end{array}$$
subject to the following initial and boundary conditions
$$\begin{array}{cc} \; & u\left(0\right)=u_{0},\;u\left(\tau \right)=m_{1}\int_{0}^{\tau }s\left(t\right)u\left(t\right)dt.\\ \; & v\left(0\right)=v_{0},\;v\left(\tau \right)=m_{2}\int_{0}^{\tau }r\left(t\right)v\left(t\right)dt. \end{array}$$
Our aim is to derive an upper bound for the proposed method. We have to calculate $R_{M}^{u}$ and $R_{M}^{v}$, defined as
(33)$$\begin{array}{cc} \; & R_{M}^{u}={|}^{c}D^{\sigma }u\left(t\right)-K_{M}G_{M\times M}^{\tau ,\sigma }\Lambda_{M}^{\tau }\left(t\right)|,\\ \; & R_{M}^{v}={|}^{c}D^{\sigma }v\left(t\right)-L_{M}G_{M\times M}^{\tau ,\sigma }\Lambda_{M}^{\tau }\left(t\right)|. \end{array}$$
We know from (Equation 14) that
$$\begin{array}{cc} \; & u\left(t\right)=K_{M}E_{M\times M}\Lambda_{M}^{\tau }\left(t\right)+{F_{1}}_{M}\Lambda_{M}^{\tau }\left(t\right)+\sum_{k=m+1}^{\infty }u_{k}\sum_{i=0}^{m}\Theta_{i,k,\tau }\rho_{i}^{\tau }\left(t\right),\\ \; & v\left(t\right)=L_{M}E_{M\times M}\Lambda_{M}^{\tau }\left(t\right)+{F_{2}}_{M}\Lambda_{M}^{\tau }\left(t\right)+\sum_{k=m+1}^{\infty }v_{k}\sum_{i=0}^{m}\Theta_{i,k,\tau }\rho_{i}^{\tau }\left(t\right), \end{array}$$
Assume, $\hat{X}=K_{M}E_{M\times M}+F_{1}$ and $\hat{Y}=L_{M}E_{M\times M}+F_{2}.$ Using Lemma (3) and Corollary (2), we can write
(34)$$\begin{array}{cc} \; & c_{l}\left(t\right)u{\left(t\right)}^{\left(\gamma \right)}={\hat{X}}_{M}^{T}B_{c_{l}}^{\gamma }\Lambda_{M}^{\tau }\left(t\right)+\sum_{k=m+1}^{\infty }u_{k}\sum_{i=0}^{m}\sum_{i^{\prime }=0}^{m}\Theta_{i,k,\tau }\overbrace{\Theta_{k,i^{\prime },\tau }^{\left(l\right)}}\rho_{i^{\prime }}^{\tau }\left(t\right),\\ \; & d_{l}\left(t\right)v{\left(t\right)}^{\left(\gamma \right)}={\hat{Y}}_{M}^{T}B_{d_{l}}^{\gamma }\Lambda_{M}^{\tau }\left(t\right)+\sum_{k=m+1}^{\infty }v_{k}\sum_{i=0}^{m}\sum_{i^{\prime }=0}^{m}\Theta_{i,k,\tau }\overbrace{\Theta_{k,i^{\prime },\tau }^{\left(l\right)}}\rho_{i^{\prime }}^{\tau }\left(t\right). \end{array}$$
Approximating $h\left(t\right)=D_{1}\Lambda_{M}^{\tau }\left(t\right)+\sum_{k=m+1}^{\infty }d_{k}\rho_{k}^{\tau }\left(t\right)$ and $g\left(t\right)=D_{2}\Lambda_{M}^{\tau }\left(t\right)+\sum_{k=m+1}^{\infty }$ $d_{k}^{\prime }\rho_{k}^{\tau }\left(t\right)$ and using (Equation 30) in (Equation 25) we get
$$\begin{array}{cc} \; & K_{M}\Lambda_{M}^{\tau }\left(t\right)-{\hat{X}}_{M}^{T}B_{c_{0}}^{0}\Lambda_{M}^{\tau }\left(t\right)-{\hat{Y}}_{M}^{T}B_{c_{1}}^{0}\Lambda_{M}^{\tau }\left(t\right)-{\hat{X}}_{M}^{T}B_{c_{2}}^{\gamma }\Lambda_{M}^{\tau }\left(t\right)-{\hat{Y}}_{M}^{T}B_{c_{3}}^{\gamma }\Lambda_{M}^{\tau }\left(t\right)-D_{1}\Lambda_{M}^{\tau }\left(t\right)=R_{M}^{u}\left(t\right),\\ \; & L_{M}\Lambda_{M}^{\tau }\left(t\right)-{\hat{X}}_{M}^{T}B_{d_{0}}^{0}\Lambda_{M}^{\tau }\left(t\right)-{\hat{Y}}_{M}^{T}B_{d_{1}}^{0}\Lambda_{M}^{\tau }\left(t\right)-{\hat{X}}_{M}^{T}B_{d_{2}}^{\gamma }\Lambda_{M}^{\tau }\left(t\right)-{\hat{Y}}_{M}^{T}B_{d_{3}}^{\gamma }\Lambda_{M}^{\tau }\left(t\right)-D_{2}\Lambda_{M}^{\tau }\left(t\right)=R_{M}^{v}\left(t\right). \end{array}$$
where $R_{M}^{u}\left(t\right)$ and $R_{M}^{v}\left(t\right)$ is defined by the relation
$$\begin{array}{cc} R_{M}^{u}\left(t\right)= & \sum_{k=m+1}^{\infty }u_{k}\rho_{k}^{\tau }\left(t\right)+\sum_{k=m+1}^{\infty }u_{k}\sum_{i=0}^{m}\sum_{i^{\prime }=0}^{m}\Theta_{i,k,\tau }\overbrace{\Theta_{k,i^{\prime },\tau }^{\left(0\right)}}\rho_{i^{\prime }}^{\tau }\left(t\right)+\sum_{k=m+1}^{\infty }v_{k}\sum_{i=0}^{m}\sum_{i^{\prime }=0}^{m}\Theta_{i,k,\tau }\overbrace{\Theta_{k,i^{\prime },\tau }^{\left(0\right)}}\rho_{i^{\prime }}^{\tau }\left(t\right)+\\ & \sum_{k=m+1}^{\infty }u_{k}\sum_{i=0}^{m}\sum_{i^{\prime }=0}^{m}\Theta_{i,k,\tau }\overbrace{\Theta_{k,i^{\prime },\tau }^{\left(1\right)}}\rho_{i^{\prime }}^{\tau }\left(t\right)+\sum_{k=m+1}^{\infty }v_{k}\sum_{i=0}^{m}\sum_{i^{\prime }=0}^{m}\Theta_{i,k,\tau }\overbrace{\Theta_{k,i^{\prime },\tau }^{\left(1\right)}}\rho_{i^{\prime }}^{\tau }\left(t\right)+\sum_{k=m+1}^{\infty }d_{k}\rho_{k}^{\tau }\left(t\right),\\ R_{M}^{v}\left(t\right)= & \sum_{k=m+1}^{\infty }v_{k}\rho_{k}^{\tau }\left(t\right)+\sum_{k=m+1}^{\infty }u_{k}\sum_{i=0}^{m}\sum_{i^{\prime }=0}^{m}\Theta_{i,k,\tau }\overbrace{\Theta_{k,i^{\prime },\tau }^{\left(0\right)}}\rho_{i^{\prime }}^{\tau }\left(t\right)+\sum_{k=m+1}^{\infty }v_{k}\sum_{i=0}^{m}\sum_{i^{\prime }=0}^{m}\Theta_{i,k,\tau }\overbrace{\Theta_{k,i^{\prime },\tau }^{\left(0\right)}}\rho_{i^{\prime }}^{\tau }\left(t\right)+\\ & \sum_{k=m+1}^{\infty }u_{k}\sum_{i=0}^{m}\sum_{i^{\prime }=0}^{m}\Theta_{i,k,\tau }\overbrace{\Theta_{k,i^{\prime },\tau }^{\left(1\right)}}\rho_{i^{\prime }}^{\tau }\left(t\right)+\sum_{k=m+1}^{\infty }v_{k}\sum_{i=0}^{m}\sum_{i^{\prime }=0}^{m}\Theta_{i,k,\tau }\overbrace{\Theta_{k,i^{\prime },\tau }^{\left(1\right)}}\rho_{i^{\prime }}^{\tau }\left(t\right)+\sum_{k=m+1}^{\infty }d_{k}^{\prime }\rho_{k}^{\tau }\left(t\right), \end{array}$$
Using the bounded property of the Legendre polynomial, it follows that
(35)$$\begin{array}{cc} |R_{M}^{u}\left(t\right)|\leq & \sum_{k=m+1}^{\infty }|u_{k}\left|\right|\left[1+\sum_{i=0}^{m}\sum_{i^{\prime }=0}^{m}\Theta_{i,k,\tau }\overbrace{\Theta_{k,i^{\prime },\tau }^{\left(0\right)}}+\sum_{i=0}^{m}\sum_{i^{\prime }=0}^{m}\Theta_{i,k,\tau }\overbrace{\Theta_{k,i^{\prime },\tau }^{\left(1\right)}}\right]|+\\ & \sum_{k=m+1}^{\infty }|v_{k}\left|\right|\left[\sum_{i=0}^{m}\sum_{i^{\prime }=0}^{m}\Theta_{i,k,\tau }\overbrace{\Theta_{k,i^{\prime },\tau }^{\left(0\right)}}+\sum_{i=0}^{m}\sum_{i^{\prime }=0}^{m}\Theta_{i,k,\tau }\overbrace{\Theta_{k,i^{\prime },\tau }^{\left(1\right)}}\right]|+\sum_{k=m+1}^{\infty }\left|d_{k}\right|,\\ |R_{M}^{v}\left(t\right)|\leq & \sum_{k=m+1}^{\infty }|v_{k}\left|\right|\left[1+\sum_{i=0}^{m}\sum_{i^{\prime }=0}^{m}\Theta_{i,k,\tau }\overbrace{\Theta_{k,i^{\prime },\tau }^{\left(0\right)}}+\sum_{i=0}^{m}\sum_{i^{\prime }=0}^{m}\Theta_{i,k,\tau }\overbrace{\Theta_{k,i^{\prime },\tau }^{\left(1\right)}}\right]|+\\ & \sum_{k=m+1}^{\infty }|u_{k}\left|\right|\left[\sum_{i=0}^{m}\sum_{i^{\prime }=0}^{m}\Theta_{i,k,\tau }\overbrace{\Theta_{k,i^{\prime },\tau }^{\left(0\right)}}+\sum_{i=0}^{m}\sum_{i^{\prime }=0}^{m}\Theta_{i,k,\tau }\overbrace{\Theta_{k,i^{\prime },\tau }^{\left(1\right)}}\right]|+\sum_{k=m+1}^{\infty }\left|d_{k}^{\prime }\right|. \end{array}$$
The above equation establishes an error bound for the solution u(t) and v(t). It also ensures the convergence of the proposed method for the solution of coupled system of FDEs.

## 6. Test Problems

We solve one single equation, three systems of linear FDEs with variable equations, and two systems of nonlinear problems, and analyze the convergence of the approximate solution by measuring the following error norms.
$$\left|\right|E_{u}{\left|\right|}_{2}=\frac{1}{\tau }\int_{0}^{\tau }U_{M}\left(t\right)-u\left(t\right)dt$$
and
$$\left|\right|E_{u}{\left|\right|}_{max}=max_{x\in [0,\tau ]}\left|U_{M}\left(t\right)-u\left(t\right)\right|.$$
We check the accuracy of the boundary condition by measuring the following error norms.
$$\left|\right|E_{u}{\left|\right|}_{b}=\left|U_{M}\left(\tau \right)-m_{1}\int_{0}^{\tau }U_{M}\left(t\right)\right|.$$
In the above bounds, the quantity $U_{M}\left(t\right)$ represents the m− term approximation to the solution u(t).

**Test** **Problem** **1.**$$\begin{array}{cc} {}^{c}D^{1.2}u\left(t\right)=\left(t^{2}+sin\left(t\right)\right)u\left(t\right) & +\left(2t-t^{3}\right)u^{0.7}\left(t\right)+f\left(t\right),\\ u(0)=4,\; & u\left(1\right)=1.1216\int_{0}^{1}cos\left(t\right)u\left(t\right)dt, \end{array}$$
where the exact solution $u\left(t\right)=t^{3}+t^{2}+t+4$, and the source term $f\left(t\right)=\frac{38683084397149375\;t^{\frac{4}{5}}\;\left(50\;t^{2}-35\;t+21\right)}{378302368699121664}-\left(sin\;\left(t\right)+t^{2}\right)\;\left(t^{4}-t^{3}+t^{2}+4\right)-\frac{150543064388819875\;t^{\frac{13}{10}}\;\left(2\;t-t^{3}\right)\;\left(400\;t^{2}-330\;t+253\right)}{22218508761632342016}.$

**Test** **Problem** **2.**$$\begin{array}{cc} {}^{c}D^{\sigma }u\left(t\right)=\left(t+1\right)u\left(t\right) & +\left(1-t\right)u^{\sigma_{1}}\left(t\right)+\left(2t\right)v\left(t\right)\\ & +\left(t^{2}\right)v^{\prime }\left(t\right)+f\left(t\right),\\ {}^{c}D^{\sigma }v\left(t\right)=\left(t^{2}+1\right)u\left(t\right) & +\left(1-t^{2}\right)u^{\sigma_{1}}\left(t\right)+\left(3t\right)v\left(t\right)\\ & +\left(t^{3}\right)v^{\prime }\left(t\right)+g\left(t\right),\\ u(0)=1,\; & u\left(1\right)=2.1270\int_{0}^{1}sin\left(t\right)u\left(t\right)dt,\\ v(0)=-1,\; & v\left(1\right)=-1.8925\int_{0}^{1}cos\left(t\right)v\left(t\right)dt. \end{array}$$
where the exact solution $u\left(t\right)=t^{2}+t^{3}+e^{t}$ and $v\left(t\right)=t^{2}-t^{3}-e^{t}$, and the source term $f\left(t\right)=6\;t+e^{t}+2\;t\;\left(e^{t}-t^{2}+t^{3}\right)-\left(t+1\right)\;\left(e^{t}+t^{2}+t^{3}\right)+\left(t-1\right)\;\left(2\;t+e^{t}+3\;t^{2}\right)+t^{2}\;\left(e^{t}-2\;t+3\;t^{2}\right)+2$ and $g\left(t\right)=3\;t\;\left(e^{t}-t^{2}+t^{3}\right)-e^{t}-6\;t+t^{3}\;\left(e^{t}-2\;t+3\;t^{2}\right)-\left(t^{2}+1\right)\;\left(e^{t}+t^{2}+t^{3}\right)+\left(t^{2}-1\right)\left(2\;t+e^{t}+3\;t^{2}\right)+2$.

**Test** **Problem** **3.**$$\begin{array}{cc} {}^{c}D^{\sigma }u\left(t\right)= & sin\left(t\right)u\left(t\right)+cos\left(t\right)u^{\sigma_{1}}\left(t\right)+\\ & \left(sin\left(t\right)+cos\left(t\right)\right)v\left(t\right)+sin\left(2t\right)v^{\prime }\left(t\right)+f\left(t\right),\\ {}^{c}D^{\sigma }v\left(t\right)= & \left(cos\left(2t\right)\right))u\left(t\right)+\left(tsin\left(t\right)\right)u^{\sigma_{1}}\left(t\right)\\ & +\left(tcos\left(t\right)\right)v\left(t\right)+\left(t^{2}sin\left(t\right)\right)v^{\prime }\left(t\right)+g\left(t\right),\\ u(0)=0,\; & u\left(1\right)=0.8\int_{0}^{1}e^{2t}sin\left(t\right)u\left(t\right)dt,\\ v(0)=1,\; & v\left(1\right)=0.71\int_{0}^{1}e^{-2t}cos\left(t\right)v\left(t\right)dt. \end{array}$$
where the exact solution $u\left(t\right)=e^{t}sin\left(t\right)$ and $v\left(t\right)=e^{-t}cos\left(t\right)$, and the source term $f\left(t\right)=2\;e^{t}\;cos\;\left(t\right)-e^{t}\;{sin\;\left(t\right)}^{2}-cos\;\left(t\right)\;\left(e^{t}\;cos\;\left(t\right)+e^{t}\;sin\;\left(t\right)\right)+sin\;\left(2\;t\right)\;\left(e^{-t}\;cos\;\left(t\right)+e^{-t}\;sin\;\left(t\right)\right)-e^{-t}\;cos\;\left(t\right)\;\left(cos\;\left(t\right)+sin\;\left(t\right)\right)$ and $g\left(t\right)=2\;e^{-t}\;sin\;\left(t\right)-t\;e^{-t}\;{cos\;\left(t\right)}^{2}-t\;sin\;\left(t\right)\;\left(e^{t}\;cos\;\left(t\right)+e^{t}\;sin\;\left(t\right)\right)-cos\;\left(2\;t\right)\;e^{t}\;sin\;\left(t\right)+t^{2}\;sin\;\left(t\right)\;\left(e^{-t}\;cos\;\left(t\right)+e^{-t}\;sin\;\left(t\right)\right).$

**Test** **Problem** **4.**$$\begin{array}{cc} {}^{c}D^{\sigma }u\left(t\right)=e^{\left(t\right)}u\left(t\right) & +e^{\left(-t\right)}u^{\sigma_{1}}\left(t\right)+\\ & \left(e^{\left(t\right)}+e^{\left(-t\right)}\right)v\left(t\right)+e^{\left(2t\right)}v^{\prime }\left(t\right)+f\left(t\right),\\ {}^{c}D^{\sigma }v\left(t\right)=\left(e^{\left(-2t\right)}\right) & u\left(t\right)+\left(te^{\left(t\right)}\right)u^{\sigma_{1}}\left(t\right)\\ & +\left(te^{\left(t\right)}\right)v\left(t\right)+\left(t^{2}e^{\left(t\right)}\right)v^{\prime }\left(t\right)+g\left(t\right),\\ u(0)=1,\; & u\left(1\right)=0.5\int_{0}^{1}\left(1-t\right)e^{2t}u\left(t\right)dt,\\ v(0)=1,\; & v\left(1\right)=10\int_{0}^{1}\left(1-t^{2}\right)e^{-2t}v\left(t\right)dt. \end{array}$$
where the exact solution $u\left(t\right)=t^{4}{\left(2-t\right)}^{3}$ and $v\left(t\right)=\left(t^{3}\right)\left(3-t\right)$, and the source term $f\left(t\right)=e^{-t}\;\left(4\;t^{3}\;{\left(t-2\right)}^{3}+3\;t^{4}\;{\left(t-2\right)}^{2}\right)-12\;t^{2}\;{\left(t-2\right)}^{3}-24\;t^{3}\;{\left(t-2\right)}^{2}-e^{2\;t}\;\left(t^{3}\;\left(2\;t-6\right)+3\;t^{2}\;{\left(t-3\right)}^{2}\right)-3\;t^{4}\;\left(2\;t-4\right)+t^{4}\;e^{t}\;{\left(t-2\right)}^{3}-t^{3}\;\left(e^{-t}+e^{t}\right)\;{\left(t-3\right)}^{2}$ and $g\left(t\right)=6\;t\;{\left(t-3\right)}^{2}+6\;t^{2}\;\left(2\;t-6\right)+2\;t^{3}-t^{4}\;e^{t}\;{\left(t-3\right)}^{2}-t^{2}\;sin\;\left(t\right)\;\left(t^{3}\;\left(2\;t-6\right)+3\;t^{2}\;{\left(t-3\right)}^{2}\right)+t^{4}\;e^{-2\;t}\;{\left(t-2\right)}^{3}+t\;e^{t}\;(4\;t^{3}\;{\left(t-2\right)}^{3}+3\;t^{4}\;{\left(t-2\right)}^{2}).$

**Test** **Problem** **5.**$$\begin{array}{cc} \; & {}^{c}D^{\sigma }u\left(t\right)=u\left(t\right)+v\left(t\right)+u^{2}\left(t\right)+v^{\prime }\left(t\right)u^{\prime }\left(t\right)+f\left(t\right),\\ \; & {}^{c}D^{\sigma }v\left(t\right)=u^{\prime }\left(t\right)+v^{\prime }\left(t\right)+v{\left(t\right)}^{2}+u\left(t\right)u^{\prime }\left(t\right)+g\left(t\right),\\ \; & u\left(0\right)=0,\;u\left(1\right)=5.4069\int_{0}^{1}sin\left(t\right)u\left(t\right)dt,\\ \; & v(0)=1,\;v(1)=0. \end{array}$$
where the exact solution $u\left(t\right)=\left(t^{2}+1\right)t^{2}$ and $v\left(t\right)=t^{3}\left(1-t\right)$, and the source term $f\left(t\right)=t^{3}\;\left(t-1\right)+\left(3\;t^{2}\;\left(t-1\right)+t^{3}\right)\;\left(2\;t\;\left(t^{2}+1\right)+2\;t^{3}\right)-t^{2}\;\left(t^{2}+1\right)-t^{4}\;{\left(t^{2}+1\right)}^{2}+12\;t^{2}+2$ and $g\left(t\right)=3\;t^{2}\;\left(t-1\right)-2\;t\;\left(t^{2}+1\right)-t^{6}\;{\left(t-1\right)}^{2}-6\;t\;\left(t-1\right)-6\;t^{2}-t^{3}-t^{2}\;\left(t^{2}+1\right)\;\left(2\;t\;\left(t^{2}+1\right)+2\;t^{3}\right).$

**Test** **Problem** **6.**$$\begin{array}{cc} {}^{c}D^{\sigma }u\left(t\right)= & \left(t+1\right)u\left(t\right)+\left(1-t\right)u^{\prime }\left(t\right)+2tv\left(t\right)\\ & +t^{2}v^{\prime }\left(t\right)+u{\left(t\right)}^{2}+v^{\prime }\left(t\right)u^{\prime }\left(t\right)-v^{3}\left(t\right)+f\left(t\right),\\ {}^{c}D^{\sigma }v\left(t\right)= & \left(t^{2}+1\right)u\left(t\right)+\left(1-t^{2}\right)u^{\prime }\left(t\right)+3tv\left(t\right)\\ & +t^{3}v^{\prime }\left(t\right)+v{\left(t\right)}^{2}+u\left(t\right)u^{\prime }\left(t\right)-u^{4}\left(t\right)+g\left(t\right),\\ u(0)=0,\; & u\left(1\right)=0.6173\int_{0}^{1}\left(2t+1\right)u\left(t\right)dt,\\ v(0)=1,\; & v\left(1\right)=0.7311\int_{0}^{1}\left(3t+1\right)v\left(t\right)dt. \end{array}$$
where the exact solution u(t)=sin(t) and $v\left(t\right)=e^{t}$, and the source term $f\left(t\right)=e^{3\;t}-sin\;\left(t\right)+cos\;\left(t\right)\;\left(t-1\right)-t^{2}\;e^{t}-e^{t}\;cos\;\left(t\right)-sin\;\left(t\right)\;\left(t+1\right)-{sin\;\left(t\right)}^{2}-2\;t\;e^{t}$ and $g\left(t\right)=e^{t}-e^{2\;t}-t^{3}\;e^{t}-cos\;\left(t\right)\;sin\;\left(t\right)+{\sin\;\left(t\right)}^{4}+cos\;\left(t\right)\;\left(t^{2}-1\right)-sin\;\left(t\right)\;\left(t^{2}+1\right)-3\;t\;e^{t}.$

## 7. Results and Discussion

The first test problem is solved using the proposed method. The problem is linear and is relatively easy to solve. We compare the approximate solution with the exact solution of the problem. We observe that by increasing the scale level of approximation, the approximate solution draws closer to the exact solution as expected; see for example Figure 1a. At M=8 the approximate solution (black line) coincides with the exact solution (red dots). In the second part of the same figure, we plot the absolute difference of the exact and approximate solution considering different scales. It is observed that at M=11, the absolute error is less than $10^{-9}$. This means accuracy up to the ninth decimal place is achieved. We also calculate all the three error norms at different scales. The results are displayed in Table 1. One can easily note that at M=5, the value of is $\left|\right|E_{u}{\left|\right|}_{2}=0.1314\times 10^{-1}$, $\left|\right|E_{u}{\left|\right|}_{max}=0.6431\times 10^{-1}$ and $\left|\right|E_{u}{\left|\right|}_{b}=8.0032\times 10^{-17}$. While increasing the scale levels, these values start to decrease with great speed. At M=13, these values become $4.5147\times 10^{-10},$
$2.4318\times 10^{-9}$ and $2.4362\times 10^{-17}$, respectively.

The analysis of the second problem is given in Table 2. We fix the orders of equation to σ=1.8 and γ=0.8 and solve the problem using scale level ranges from M=5 to M=13. We observe that the error norm decreases rapidly with the increase in scale level. For example, the value $\left|\right|E_{u}{\left|\right|}_{2}$ at M=5 is $1.1044\times 10^{-1}$ and at M=13, this value drops to $4.5177\times 10^{-10},$ which is very high accuracy. At the same scale, the value of $\left|\right|E_{v}{\left|\right|}_{2}$ is $1.3237\times 10^{-10}$. The value of norms $\left|\right|E_{u}{\left|\right|}_{max}$ and $\left|\right|E_{v}{\left|\right|}_{max}$ are $1.4501\times 10^{-9}$ and $2.2449\times 10^{-10}$, respectively. Similarly the approximate solution satisfy the boundary condition with high accuracy. The errors in the boundary condition are $9.7194\times 10^{-17}$ and $6.2399\times 10^{-16}$. The error in the boundary condition is observed to be constant, that is, we have the same accuracy at all scale level. The accuracy of the proposed method for all possible values of $\sigma_{1}$ and $\sigma_{2}$ are analyzed by calculating the error norm $\left|\right|E_{v}{\left|\right|}_{2}$. The error norm for the solution u(t) and v(t) are displayed in Figure 2. We observe that the method produces excellent approximation to the solution at almost all values of parameters.

The same analysis is performed for Test Problem 3 and Test Problem 4. The results are shown in Table 3 and Table 4. The same conclusion is reached for these two examples. The error bounds are shown in Figure 3 and Figure 4. We observe that the method yields an almost accurate solution for all values of these parameters. In Figure 2, we can easily see that the error norm is less than $10^{-3}$. Note that for these problems we have set M=5. It is always possible to get a more accurate solution by selecting the highest choices of scale level.

The nonlinear Test Problem 5 is solved with the proposed iterative scheme. We use three different choices of the parameters σ and γ and calculate the the error norms $\left|\right|E_{u}{\left|\right|}_{2}$ at different iterations. We use seven iterations for the approximation of solution. The results are displayed in Figure 5 and Figure 6. We observe that the error decreases with respect to the number of iterations and is highly convergent. Note that for this example, we fix the scale level M=10. It is clear form the figure that the proposed method is highly efficient, especially in the solution of nonlinear equations. The same phenomenon is observed for Test Problem 6. The results are displayed in Figure 7.

## 8. Conclusions and Future Work

This article presents a new algorithm for the solution of fractional order differential equations. The Newton Raphson method is combined with the operational matrix method for the solution of these problems. The convergence of the proposed method is checked analytically and is confirmed by solving several test problems. It is found that the approximate solution is highly accurate, and one can get high accuracy by using high scale levels. The mathematical proof of convergence and error analysis is our future plan of research.

## Figures and Tables

**Figure 1 entropy-23-01154-f001:**
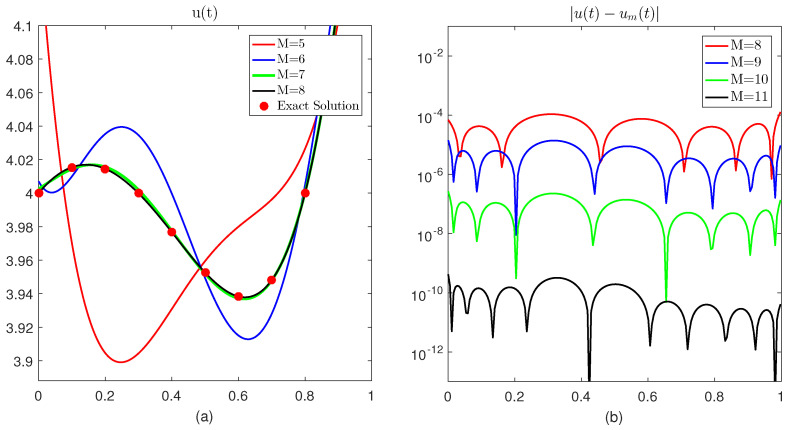
(**a**) Comparison of exact and approximate solution at different scale levels of Test Problem 1. (**b**) Absolute difference in the exact and approximate solutions at different scale levels of Test Problem 1.

**Figure 2 entropy-23-01154-f002:**
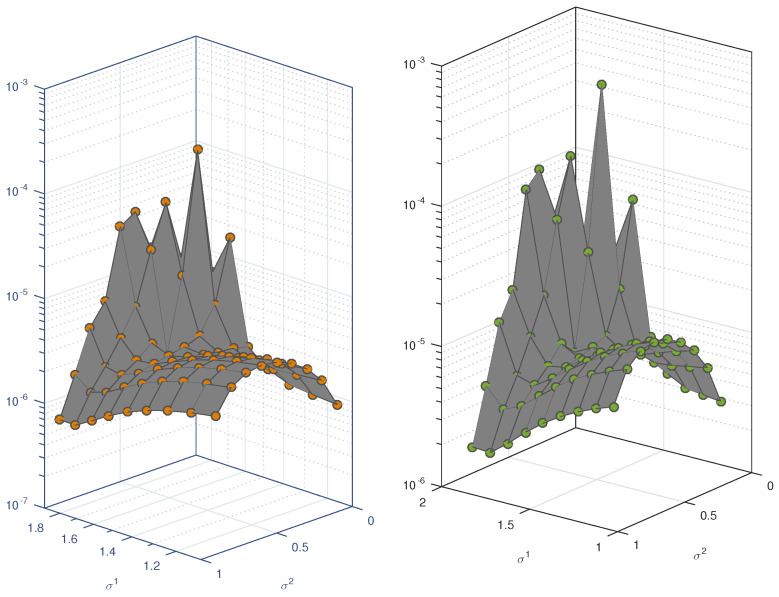
Errornorm for different values of $\sigma^{1}$ and $\sigma^{2}$ in Test Problem 2.

**Figure 3 entropy-23-01154-f003:**
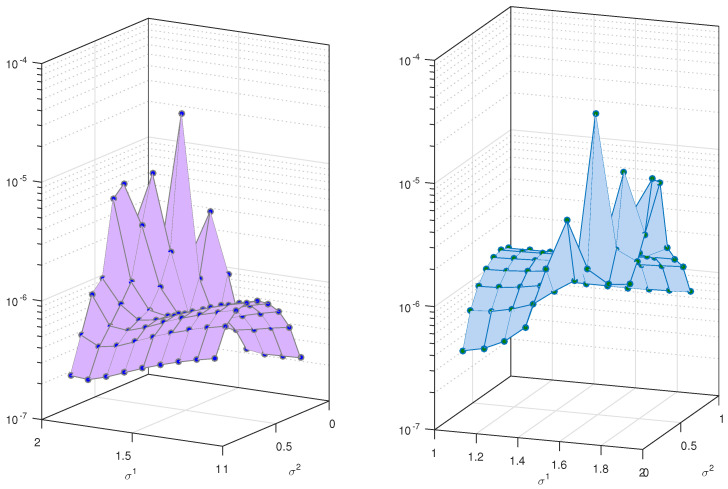
Errornorm for different values of $\sigma^{1}$ and $\sigma^{2}$ in Test Problem 3.

**Figure 4 entropy-23-01154-f004:**
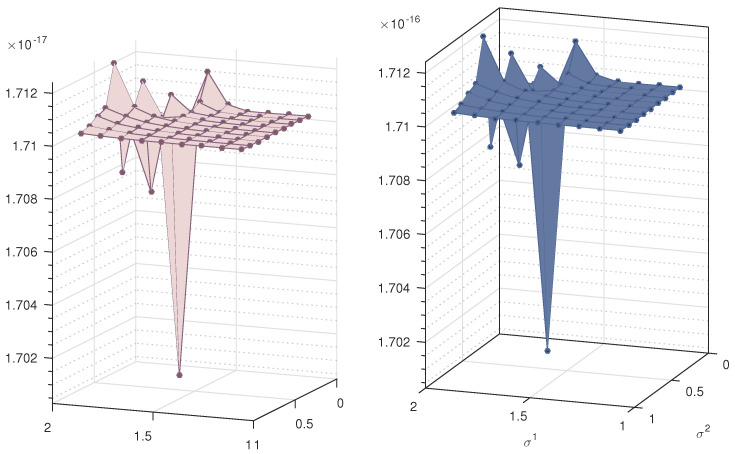
Error norm on boundary for different values of $\sigma^{1}$ and $\sigma^{2}$ in Test Problem 4.

**Figure 5 entropy-23-01154-f005:**
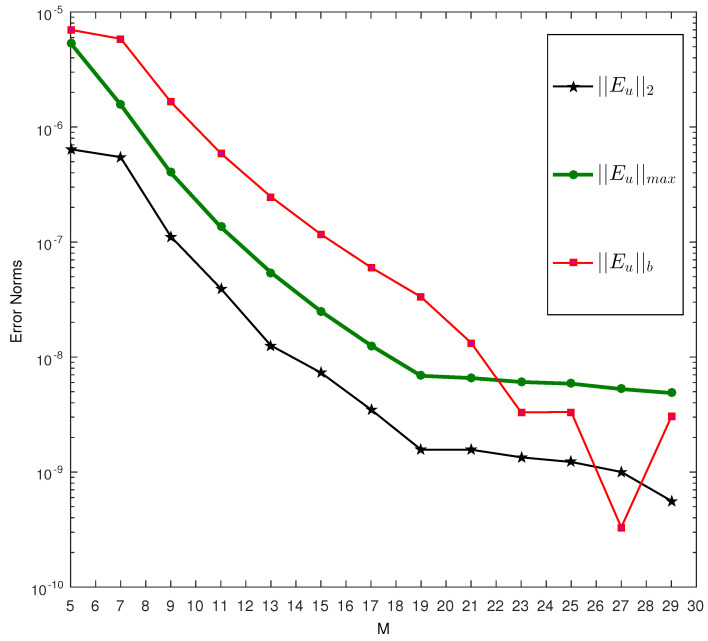
Error norm of Test Problem 4.

**Figure 6 entropy-23-01154-f006:**
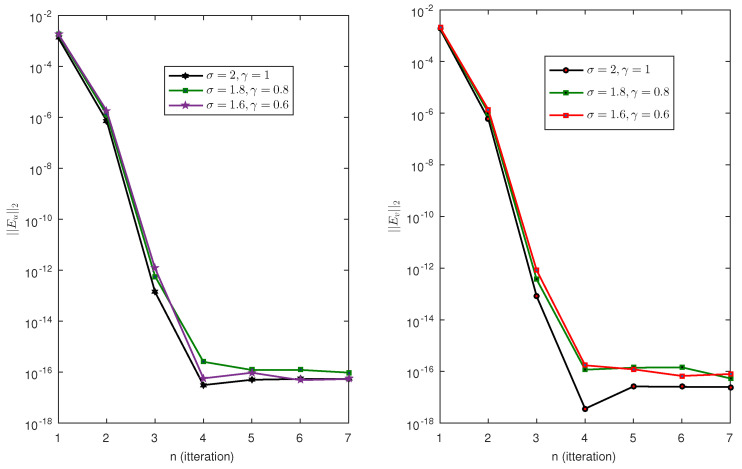
Error norm of Test Problem 5.

**Figure 7 entropy-23-01154-f007:**
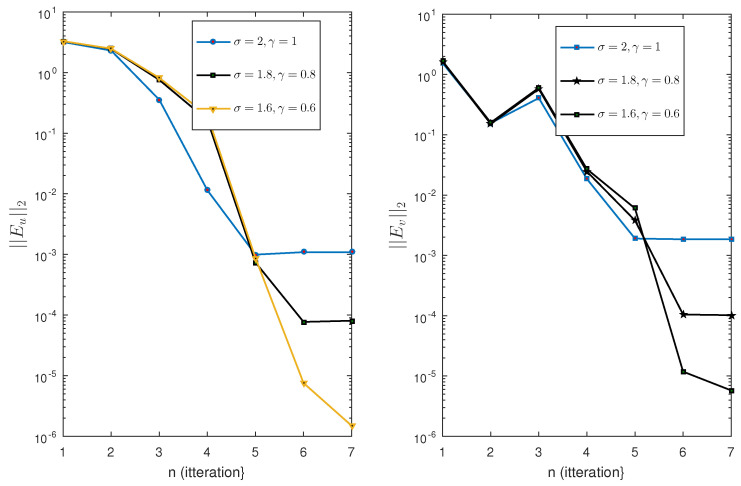
Error norm of Test Problem 6.

**Table 1 entropy-23-01154-t001:** Error norms of Test Problem 1 at scale level M=5:13.

M	$\left|\right|E_{u}{\left|\right|}_{2}$	$\left|\right|E_{u}{\left|\right|}_{\max}$	$\left|\right|E_{u}{\left|\right|}_{b}$
5	$0.1314\times 10^{-1}$	$0.6431\times 10^{-1}$	$8.0032\times 10^{-17}$
6	$1.1355\times 10^{-2}$	$9.3415\times 10^{-1}$	$0.4367\times 10^{-16}$
7	$2.3631\times 10^{-3}$	$8.3640\times 10^{-3}$	$3.0669\times 10^{-16}$
8	$3.9474\times 10^{-5}$	$3.6641\times 10^{-4}$	$9.2291\times 10^{-17}$
9	$1.5963\times 10^{-5}$	$0.5933\times 10^{-5}$	$8.3432\times 10^{-17}$
10	$1.6057\times 10^{-6}$	$0.5364\times 10^{-6}$	$7.7554\times 10^{-17}$
11	$1.5823\times 10^{-7}$	$1.0592\times 10^{-7}$	$6.9857\times 10^{-17}$
12	$7.4855\times 10^{-9}$	$9.0092\times 10^{-8}$	$5.4743\times 10^{-17}$
13	$4.5147\times 10^{-10}$	$2.4318\times 10^{-9}$	$2.4362\times 10^{-17}$

**Table 2 entropy-23-01154-t002:** Error norms of Test Problem 2 at scale level M=5:13.

M	$\left|\right|E_{u}{\left|\right|}_{2}$	$\left|\right|E_{v}{\left|\right|}_{2}$	$\left|\right|E_{u}{\left|\right|}_{\max}$	$\left|\right|E_{v}{\left|\right|}_{\max}$	$\left|\right|E_{u}{\left|\right|}_{b}$	$\left|\right|E_{v}{\left|\right|}_{b}$
5	$1.1044\times 10^{-1}$	$4.3598\times 10^{-2}$	$3.6499\times 10^{-1}$	$6.5235\times 10^{-2}$	$7.9467\times 10^{-17}$	$6.2402\times 10^{-16}$
6	$1.9005\times 10^{-2}$	$9.3636\times 10^{-3}$	$1.6085\times 10^{-1}$	$1.6796\times 10^{-2}$	$9.7023\times 10^{-16}$	$6.2370\times 10^{-16}$
7	$2.3621\times 10^{-3}$	$7.6605\times 10^{-4}$	$7.5980\times 10^{-3}$	$1.6380\times 10^{-3}$	$9.7237\times 10^{-17}$	$6.2396\times 10^{-16}$
8	$3.9834\times 10^{-5}$	$4.9422\times 10^{-5}$	$1.0331\times 10^{-4}$	$1.0113\times 10^{-4}$	$9.7191\times 10^{-17}$	$6.2400\times 10^{-16}$
9	$1.4293\times 10^{-5}$	$5.3436\times 10^{-6}$	$4.5571\times 10^{-5}$	$8.6999\times 10^{-6}$	$9.7195\times 10^{-17}$	$6.2400\times 10^{-16}$
10	$1.9227\times 10^{-6}$	$5.0884\times 10^{-7}$	$6.1295\times 10^{-6}$	$7.3992\times 10^{-7}$	$9.7194\times 10^{-17}$	$6.2399\times 10^{-16}$
11	$1.2515\times 10^{-7}$	$3.1891\times 10^{-8}$	$4.0112\times 10^{-7}$	$6.2640\times 10^{-8}$	$9.7194\times 10^{-17}$	$6.2399\times 10^{-16}$
12	$7.1330\times 10^{-9}$	$1.9583\times 10^{-9}$	$2.2596\times 10^{-8}$	$3.6441\times 10^{-9}$	$9.7194\times 10^{-17}$	$6.2399\times 10^{-16}$
13	$4.5177\times 10^{-10}$	$1.3237\times 10^{-10}$	$1.4501\times 10^{-9}$	$2.2449\times 10^{-10}$	$9.7194\times 10^{-17}$	$6.2399\times 10^{-16}$

**Table 3 entropy-23-01154-t003:** Error norms of Test Problem 3 at scale level M=5:13.

N	$\left|\right|E_{u}{\left|\right|}_{2}$	$\left|\right|E_{v}{\left|\right|}_{2}$	$\left|\right|E_{u}{\left|\right|}_{\max}$	$\left|\right|E_{v}{\left|\right|}_{\max}$	$\left|\right|E_{u}{\left|\right|}_{b}$	$\left|\right|E_{v}{\left|\right|}_{b}$
5	$6.5505\times 10^{-3}$	$5.1204\times 10^{-4}$	$1.5519\times 10^{-2}$	$8.6002\times 10^{-4}$	$7.0670\times 10^{-17}$	$1.8084\times 10^{-18}$
6	$3.2583\times 10^{-4}$	$2.8746\times 10^{-5}$	$7.7139\times 10^{-4}$	$4.7281\times 10^{-5}$	$7.0336\times 10^{-17}$	$1.6297\times 10^{-18}$
7	$4.6407\times 10^{-6}$	$4.3808\times 10^{-7}$	$1.0949\times 10^{-5}$	$7.1483\times 10^{-7}$	$7.0320\times 10^{-17}$	$1.6203\times 10^{-18}$
8	$4.6870\times 10^{-7}$	$4.6599\times 10^{-8}$	$1.1065\times 10^{-6}$	$7.8519\times 10^{-8}$	$7.0320\times 10^{-17}$	$1.6201\times 10^{-18}$
9	$4.3761\times 10^{-8}$	$3.4302\times 10^{-9}$	$1.0365\times 10^{-7}$	$5.7780\times 10^{-9}$	$7.0320\times 10^{-17}$	$1.6201\times 10^{-18}$
10	$1.3030\times 10^{-9}$	$1.1448\times 10^{-10}$	$3.0748\times 10^{-9}$	$1.8688\times 10^{-10}$	$7.0320\times 10^{-17}$	$1.6201\times 10^{-18}$
11	$1.2252\times 10^{-11}$	$1.1526\times 10^{-12}$	$2.8927\times 10^{-11}$	$1.8785\times 10^{-12}$	$7.0320\times 10^{-17}$	$1.6201\times 10^{-18}$
12	$7.2561\times 10^{-13}$	$7.2734\times 10^{-14}$	$1.7118\times 10^{-12}$	$1.1828\times 10^{-13}$	$7.0320\times 10^{-17}$	$1.6201\times 10^{-18}$
13	$4.6262\times 10^{-14}$	$3.6748\times 10^{-15}$	$1.0959\times 10^{-13}$	$6.1876\times 10^{-15}$	$7.0320\times 10^{-17}$	$1.6201\times 10^{-18}$

**Table 4 entropy-23-01154-t004:** Error norms of Test Problem 4 at scale level M=5:13.

N	$\left|\right|E_{u}{\left|\right|}_{2}$	$\left|\right|E_{v}{\left|\right|}_{2}$	$\left|\right|E_{u}{\left|\right|}_{\max}$	$\left|\right|E_{v}{\left|\right|}_{\max}$	$\left|\right|E_{u}{\left|\right|}_{b}$	$\left|\right|E_{v}{\left|\right|}_{b}$
5	0.0108	0.0036249	0.030961	0.0095518	$-3.5187\times 10^{-17}$	$3.7849\times 10^{-15}$
6	0.0048582	0.0020882	0.01299	0.0075678	$3.4932\times 10^{-17}$	$3.78\times 10^{-15}$
7	0.00055459	0.00024661	0.001525	0.00089155	$3.4845\times 10^{-17}$	$3.7773\times 10^{-15}$
8	9.6413 × $10^{-16}$	$7.5388\times 10^{-16}$	$1.8345\times 10^{-15}$	$1.2652\times 10^{-15}$	$3.486\times 10^{-17}$	$3.7777\times 10^{-15}$
9	$6.8784\times 10^{-16}$	$1.4702\times 10^{-15}$	$1.3601\times 10^{-15}$	$2.8364\times 10^{-15}$	$3.4861\times 10^{-17}$	$3.7777\times 10^{-15}$
10	$4.5115\times 10^{-15}$	$5.6863\times 10^{-16}$	$1.302\times 10^{-14}$	$8.6167\times 10^{-16}$	$3.4861\times 10^{-17}$	$3.7777\times 10^{-15}$
11	$8.4757\times 10^{-17}$	$2.7414\times 10^{-16}$	$1.6027\times 10^{-16}$	$4.7286\times 10^{-16}$	$3.4861\times 10^{-17}$	$3.7777\times 10^{-15}$
12	$1.5906\times 10^{-14}$	$3.1092\times 10^{-15}$	$4.6821\times 10^{-14}$	$6.3783\times 10^{-15}$	$3.4861\times 10^{-17}$	$3.7777\times 10^{-15}$
13	$4.926\times 10^{-15}$	1.0866×10−15	$1.4378\times 10^{-14}$	$2.0953\times 10^{-15}$	$3.4861\times 10^{-17}$	$3.7777\times 10^{-15}$

## Data Availability

The data used to support the findings of this study are included within the article.

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
