# Peer review of "Extension of Operational Matrix Technique for the Solution of Nonlinear System of Caputo Fractional Differential Equations Subjected to Integral Type Boundary Constrains"

_entropy, 2021, doi:10.3390/e23091154_

Round 1
Reviewer 1 Report
Journal name: Entropy (ISSN 1099-4300)
Manuscript Number: entropy-1262476
Title: A Novel Method for Solution of Nonlinear System of Fractional Order Differential Equation Subject to Integral Type Boundary Constrains.
The report
The article is thoroughly reviewed keeping in view all kind aspects which are necessary for an article to be published in some reputable international journal. Once reviewed in detail, it seems to me, it is an excellent contribution and fulfills the pre-requisites of acceptance in any good journal. My detailed review is as follows:
The authors have focused on a very interesting and important fractional differential equation: this topic is highly useful and has a great impact for a large community of researchers.
They extended the operational matrices technique to approximate the solution of coupled system of nonlinear FDEs.
development of computational scheme. The scheme presented in this article easily convert nonlinear coupled system of FDEs subject to to an equivalent algebraic structure. The error analysis and convergence of the mechanism is illustrated through a variety of problems.
The present work successfully addresses the investigated problem, and the used technique is very effective for discussing such kind of problems. Thus, the article should be of interest to computational filed
The paper is well written in terms of language but it is better to improve the English.
Final Decision: Considering all belongings, I therefore strongly recommend its acceptance after this minor correction to your well-known journal.
Author Response
Dear Reviewer
The response is attached as pdf file.

Reviewer 2 Report
The paper is not well written. I’ll write some comments:
- The title has to be changed. Nowhere in the paper is explained what is the novelty of this method? Actually, in the paper well known method is applied to a new equation. It is not called a novelty method.
- What kind of method is applied- there are exact methods, numerical methods, approximate methods, etc.
- There different types of fractional derivatives> It has to be specified in the title, as well as in the Abstract.
- Line 4- there is no information about the country.
- What are u and v in (1) ? Are they scalar functions?
- Which fractional derivative is used in (1)? The notation of the given fractional derivative on page 2 does not coincide with this one. The same is about the example.
- What does the black square in (2) mean?
- The citation of the paper [31] is so strange on lines 61 and 62.
- On the first line of p. 5 a proof is given, but no Theorem, Lemma or proposition is set up before the proof. The same is on the other pages of the paper.
- It is written “The analysis of the first problem is given in Table 1.” Actually in Table 1 some numbers are given and they have to be explained and discussed. The same is about the other tables.
- Analyzing the convergence of the suggested method on some particular problems, as it is done in this paper is not acceptable. This does not prove the convergence of the method. It does not prove that it is a good method.
As an overall the paper is not written seriously, it is full with typos, with missing parts, etc. Additionally, this paper consists only of some calculations not theoretical investigations of the convergence, of the errors of the suggested method. Definitely, it is not acceptable for publication.
Author Response

(The authors gave the same response as above.)

Reviewer 3 Report
The authors need to respond to the following questions and comments:
1- Abstract needs to rewritten and provide clear details of used methodology and techniques. Also, the significance and novelty of this work need to be highlighted which is currently is not enough.
2- The authors need to clearly provide the definition of the proposed Lemma on page 4 line 70. In addition, the proof needs to be clear. There are 2 proofs without indicating the purpose of proof.
3- The authors need to provide proof of convergence and stability of the method in detail. The convergence only has mentioned in the conclusion section that has been studied. 5 numerical examples are not sufficient to prove the convergence of the numerical method.
Author Response

(The authors gave the same response as above.)

Reviewer 4 Report
In this paper, the authors have extended the operational matrices technique to approximate the solution of a coupled system of nonlinear fractional differential equations along with the integral type boundary conditions. The proposed method is based on orthogonal polynomials. Several matrices are derived, which have strong application in the development of computational schemes. The error analysis and convergence of the mechanism are illustrated.
Observations:
1). The coefficients I_i,k in (4), are denoted by I_(I,k) in (5). Why? Notations should be uniform.
2). Verify Eqs. (8)-(10). They are incorrect. The matrix product is not possible.
3). Verify the definition of (11). If the indexes l, m, n are the summation indexes, then these indexes cannot be free indexes in the left-hand side term.
The notations in these expressions are not correct.
4). The physical significance of the system (39) should be explained. The authors should give the possible practical applications of the studied model.
Author Response

(The authors gave the same response as above.)

Round 2
Reviewer 2 Report
I continue to think that this paper is written careless and it has not theoretical contributions neither practical applications. There are four authors of this manuscript, but the answer is written only by one author and everywhere in this answer it is written "I". Additionally, the answer is full with grammar mistakes.
The revised version is also not written with mathematical care. The added Section 4 about the convergence is about the equation (35) which is totally different than the studied ones. Also, the written at the end of this section "The mechanism for proving the convergence of the scheme for coupled system of two equations (my comment- as the studied in the paper) is analogous. Therefore we leave it as an exercise for the readers" is strange. So, the authors could leave all proofs in the paper as excises of the readers!!!
About Section 5- 6 problems are defined but nowhere tau is given. Also, not the explicit form of the function f(t) is provided neither the exact solution is given. Then I don't understand how the graph of the exact solution is plotted. Also, how is the approximate solution obtained without the knowledge of f(t)?
Definitely, this paper is NOT appropriate for publication at this very good journal.
Author Response
All the comments raised by the reviewer are incorporated.

Reviewer 3 Report
1- The revised version covers most of the requested comments, instead of the first comment on Anstract.
C1: Abstract needs to be rewritten and provide clear details of used methodology and techniques. Also, the significance and novelty of this work need to be highlighted which is currently is not enough.
2- Remove the full stop from the end of the title.
The rest of the modifications are adequate.
Round 3
Reviewer 2 Report
The authors’ answers to my second review are not adequate.
Additionally, I’ll point out some incomplete, unclear and mistaken parts in the last version to prove my negative opinion about the acceptance of this paper:
- In the entire section 2 nothing is written about the fractional orders. Are the provided results true for all order (I’m not sure). For example, in Lemma 4 nothing is written about sigma, i.e. the fractional order. Also, this Lemma is full with grammatical and punctuation errors. The same is about Corollary 3.
- The main studied system (1) is rewritten as (19) (Only some slight changes of the notations are applied). Why instead of writing (19) the system (1) is not cited?
- Section 5 is not adequate to the studied problem (1). It has to be rewritten with significant changes. I’ll point out these changes
- Test Problem 1 is about a single equation and it is useless. It has to be deleted.
- Test Problem 2 is not a partial case of (1), In (1) the first derivatives are not included but in the test problem 2 v’(t) is included. It has to be changed, v’(t) has to be deleted or (1) has to be changed. The same is about all other provided test problems 2-6.
- The power of u(t) is labeled by sigma_1 and its value is not given (on page 18 a value of gamma is given but there is no gamma in the problem). The same is the situation about all other test problems.
- In all test problems the fractional orders of both u and v are the same but the studied problem (1) is about different orders. Now all provided 5 test problems are very similar, their comparison is not discussed, and they differ (1). To illustrate the suggested scheme it is enough to give 1-2 test problems with equal fractional orders (and a power of v(t) instead of a first derivative)
- Add at least one test problem with different fractional orders.
- Also, in (1) the upper limit of the integral in boundary value condition is not tau as in the provided example, but omega_1 and omega_2. So, at least one example with an integral form 0 to omega<1 has to be provided. Otherwise, the authors have to change (1) and to lower the novelty of the paper.
- English has to be polish. It is not acceptable to have a sentence such as “The base of our approach the most simple orthogonal polynomials” (see the abstract)
- There is a contradiction between the written in the Abstract and the Conclusion. In the abstract it is written “The convergence of the proposed method is investigated analytically” but in the Conclusion the authors wrote “The mathematical proof of convergence is our future plan of research.” So, what is the truth?